# VidHal: Benchmarking Temporal Hallucinations in Vision LLMs

**Wey Yeh Choong**                                          *c.weyyeh@u.nus.edu*
*National University of Singapore*

**Yangyang Guo**[*]                                          *guoyang.eric@gmail.com*
*National University of Singapore*

**Mohan Kankanhalli**                                        *mohan@comp.nus.edu.sg*
*National University of Singapore*

**Reviewed on OpenReview:** *https://openreview.net/forum?id=7ccWCDbdM1*

## Abstract

Vision Large Language Models (VLLMs) are widely acknowledged to be prone to hallucinations. Existing research addressing this problem has primarily been confined to image inputs, with sparse exploration of their video-based counterparts. Furthermore, current evaluation methods fail to capture nuanced errors in generated responses, which are often exacerbated by the rich spatiotemporal dynamics of videos. To address these two limitations, we introduce VIDHAL, a benchmark specially designed to evaluate video-based hallucinations in VLLMs. VIDHAL is constructed by bootstrapping video instances across a wide range of common temporal aspects. A defining feature of our benchmark lies in the careful creation of captions representing varying levels of hallucination associated with each video. To enable fine-grained evaluation, we propose a novel caption ordering task requiring VLLMs to rank captions by hallucinatory extent. We conduct extensive experiments on VIDHAL and comprehensively evaluated a broad selection of models, including both open-source and proprietary ones such as GPT4.1 and Gemini 2.5. Our results uncover significant limitations in existing VLLMs regarding video-based hallucination generation. Through our benchmark, we aim to inspire further research on I) holistic understanding of VLLM capabilities, particularly regarding hallucination, and II) advancing VLLMs to alleviate this problem. Our VIDHAL dataset and evaluation code are publicly available at `https://github.com/Lookuz/VidHal`.

## 1 Introduction

Building on the advancements of Large Language Models (LLMs), Vision LLMs (VLLMs) have recently gained significant attention. Models such as LLaVA (Liu et al., 2023; 2024c) have shown impressive performance across various visual understanding tasks involving both images and videos. Despite their potential, VLLMs are notably prone to hallucinations, where generated responses appear plausible but contradict visual context (Bai et al., 2024; Xu et al., 2024). This problem significantly compromises the reliability of VLLMs, hindering their practical use in real-world applications.

To tackle this challenge, some methods propose to leverage post-hoc techniques such as contrastive decoding (Leng et al., 2024; Zhu et al., 2024c; Favero et al., 2024; Zhuang et al., 2024) and attention calibration (Huang et al., 2024; Ma et al., 2024; Liu et al., 2024f; Yue et al., 2024; Gong et al., 2024; Zhou et al., 2024a; Xing et al., 2024b). Other efforts have been devoted to the evaluation of hallucinations in VLLMs. For example, CHAIR (Rohrbach et al., 2018) initially studies object-based hallucination evaluation with the aid of the image captioning task. Subsequent studies (Li et al., 2023e; Liu et al., 2024e; Kaul et al., 2024;

---

[*]Corresponding author.

Ding et al., 2024) instead harness paired ⟨*positive, hallucinatory*⟩ questions to probe such hallucinations. Additionally, MMHalBench (Sun et al., 2024) and AMBER (Wang et al., 2023) expand beyond object-based evaluations by constructing benchmarks that cover attribute and relationship hallucinations.

Unlike their image-based counterparts, video hallucinations pose unique challenges primarily due to the intricate spatiotemporal dynamics of videos (Fu et al., 2024; Liu et al., 2024g; Ning et al., 2023). In particular, video-specific temporal aspects, such as movement direction and chronological order of events, are especially concerning for video-based VLLMs. Furthermore, the richness of video content necessitates a finer-grained understanding, making VLLMs more vulnerable to nuanced hallucinations. Nonetheless, to the best of our knowledge, video-based hallucinations remain underexplored in the existing literature.

To address this research gap, we present VIDHAL, a benchmark specifically designed to evaluate video-based hallucinations of VLLMs. VIDHAL features videos that comprehensively cover a broad range of temporal aspects, such as entity actions and sequence of events. Each video is automatically annotated with multiple captions exhibiting *varying levels* of aspect-specific hallucinations, capturing both subtle and significant discrepancies. In addition, we perform detailed human validation to ensure the robustness and reliability of our annotation process. An additional motivation stems from the limited metrics for quantifying hallucinations in VLLMs. To capture fine-grained hallucinatory errors of these models, we propose a unique caption ordering task that requires models to rank captions by hallucination levels, con-

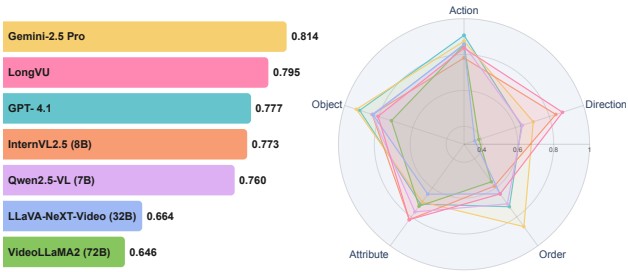

Figure 1: Multiple-Choice Question Answering performance of representative VLLMs on our VIDHAL benchmark. (Left) Ranking of representative evaluated VLLMs. (Right) Detailed accuracy results for each temporal aspect, where higher scores indicate fewer hallucinations.

sequently leading to a novel ranking-based metric and a multiple-choice question answering metric, both distinct from prior binary-accuracy based studies and specifically tailored to evaluate nuanced hallucinations in video-based VLLMs.

Using our VIDHAL dataset, we benchmark thirteen VLLMs including both open-sourced and proprietary models, with abstracted results summarized in Figure 1. Through these extensive experiments, we identify limitations in nuanced video understanding among all evaluated VLLMs. Specifically, our findings reveal that existing VLLMs struggle to differentiate between captions with varying levels of hallucination. This deficiency is particularly evident when evaluating video-specific aspects, such as *Direction* and *Order*, as illustrated in Figure 1, indicating substantial room for improvement in current video-based VLLMs.

The contributions of this work are three-fold:

- We present VIDHAL, a benchmark dataset dedicated to video-based hallucination evaluation of VLLMs. Our dataset is distinguished by i) video instances encompassing a diverse range of temporal concepts and ii) captions with varying hallucination levels.

- We introduce a novel evaluation task of caption ordering along with two metrics designed to evaluate fine-grained hallucination generation in existing VLLMs.

- We conduct extensive experiments on VIDHAL with a variety of VLLMs, uncovering limitations in their fine-grained video reasoning abilities, particularly in their tendency to generate hallucinations.

## 2 Related Work

**Vision Large Language Models.** The emergence of powerful LLMs has advanced the development of VLLMs. Typical methods in this category include LLaVA (Liu et al., 2023), MiniGPT-4 (Zhu et al., 2024a), InstructBLIP (Dai et al., 2023), and Qwen-VL (Wang et al., 2024a; Bai et al., 2025). These VLLMs rely on aligning vision encoders with LLMs using connective modules such as Q-Former (Dai et al., 2023; Zhang

et al., 2023; Cheng et al., 2024) or MLPs (Liu et al., 2024c; Su et al., 2023) with the instruction tuning stage. Recent methods have extended visual inputs from images to (long) videos, delivering impressive joint spatial-temporal reasoning capabilities. For instance, VideoLLaMA2 (Cheng et al., 2024) enhances the LLaMA model with video understanding capabilities through a Spatial-Temporal Convolution (STC) module. LLaVA-NeXT-Video (Liu et al., 2024d; Zhang et al., 2024) presents an AnyRes approach that enables reasoning with long videos.

**Hallucinations in VLLMs.** Despite their impressive performance on visual reasoning benchmarks, current VLLMs remain notoriously susceptible to hallucinations (Jiang et al., 2024; Liu et al., 2024f; Zhu et al., 2024b; Chen et al., 2024a). A common demonstration is that generated responses contain information inconsistent with the visual content (Liu et al., 2024b; Yuan et al., 2024; Xing et al., 2024a). Most approaches address the hallucination problem with post-hoc techniques. For example, LURE (Zhou et al., 2024c) and Woodpecker (Yin et al., 2023) develop pipelines that assist VLLMs in revising their responses using expert models. To reduce bias from unimodal and statistical priors, contrastive decoding methods, such as VCD (Leng et al., 2024) and M3ID (Favero et al., 2024), along with attention calibration techniques like OPERA (Huang et al., 2024) are employed to refine token predictions. Building on the success of reinforcement learning in LLM development (Ouyang et al., 2022), HA-DPO (Zhao et al., 2023), POVID (Zhou et al., 2024b) and CSR (Zhou et al., 2024d) adopt this paradigm to fine-tune VLLMs, yielding outputs with fewer hallucinations.

**Video Reasoning Benchmarks.** The rise of video-based VLLMs has driven the development of numerous video benchmarks. Notable examples, such as SEEDBench (Li et al., 2023a), VideoBench (Ning et al., 2023), MVBench (Li et al., 2024b), and VideoMME (Fu et al., 2024), focus on dynamic events requiring temporal reasoning beyond individual frames. However, these benchmarks often lack diversity in reasoning tasks and visual concepts. To address this, AutoEval-Video (Chen et al., 2023) and Perception Test (Patraucean et al., 2023) introduce complex reasoning tasks such as counterfactual and explanatory reasoning, while TempCompass (Liu et al., 2024g) expands temporal concept coverage. Several benchmarks (Li et al., 2023e; Wang et al., 2023; Sun et al., 2024; Kaul et al., 2024; Liu et al., 2024a; Wei et al., 2024; Chen et al., 2024b) have been constructed to quantify visual hallucinations, primarily targeting object-based hallucinations in images. HallusionBench (Guan et al., 2024), VideoCon (Bansal et al., 2024), and Vript (Yang et al., 2024) provide partial coverage of video-based hallucinations, while VidHalluc (Li et al., 2024a) and VideoHallucer (Wang et al., 2024b) introduce benchmarks for hallucination detection in videos. However, these benchmarks provide limited coverage of spatio-temporal concepts, focusing on conventional aspects like actions while neglecting other video-centric elements such as direction. *Additionally, their evaluation strategies primarily follow image-based approaches, which we argue are less effective in capturing nuanced, video-specific hallucinations.*

## 3 VidHal Dataset Construction

We introduce VIDHAL, a unique video-language benchmark designed to evaluate hallucinations of VLLMs in a comprehensive manner. As depicted in Figure 2, VIDHAL comprises of video instances which span a diverse spectrum of temporal aspects, including previously unexplored aspects such as directional movement. In contrast to previous studies on video hallucination evaluation (Yang et al., 2024; Wang et al., 2024b; Li et al., 2024a), VIDHAL incorporates multiple hallucinated captions per video, enabling the assessment of video hallucinations at multiple levels of granularity.

### 3.1 Temporal Hallucinations in Videos

Despite extensive studies on hallucinations in LLMs and VLLMs, the existing literature has not converged on a consensus formal definition. Following the majority of prior work in hallucination evaluation for VLLMs (Li et al., 2023e; Wang et al., 2024b; Li et al., 2024a), we treat hallucinations as a specific class of errors in which the model generates content that directly contradicts visually grounded evidence. This is distinct from errors arising from imperfect reasoning or contextual analysis that do not necessarily conflict with the visual content. Compared to images, video hallucinations extend beyond static visual elements to include misperceptions of dynamic changes within scenes. We categorize these temporal hallucinations into two semantic levels:

Figure 2: Overview of our VidHal benchmark construction pipeline. Using *direction* as an example from the five selected aspects, we begin by sourcing relevant video instances from existing datasets. Next, the anchor (positive) caption is generated from the original video metadata. Finally, GPT-4o is employed to generate hallucinatory captions at varying levels.

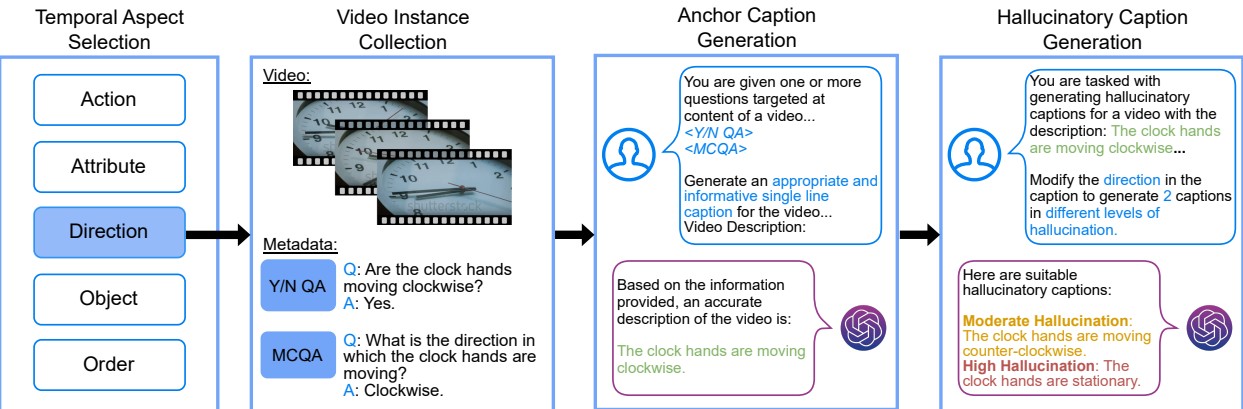

**Lexical Semantics (L-Sem)** captures instances where VLLMs misinterpret words related to temporal features, including nouns referring to objects or attributes (e.g., misidentifying a color change from green to red as green to orange) and verbs describing actions (e.g., interpreting "kicking a ball" as "throwing a ball").

**Clause Semantics (C-Sem)** encompasses errors involving event descriptions and their sequences, where the VLLM incorrectly predicts the order of events occurring in the video. For example, given sequentially occurring events $A$ and $B$ in a video, the model may perceive $B$ preceding $A$.

By addressing these two dimensions of video-based hallucinations, VidHal offers holistic coverage over the level of detail in which VLLMs may hallucinate. This two-level decomposition is grounded in the temporal ontology (Moens & Steedman, 1988), which formally establishes that both lexical and clause-level representations are necessary and sufficient for complete natural-language temporal description, spanning intra-event structure (L-Sem) through to inter-event relational ordering (C-Sem).

## 3.2 Temporal Concept Selection

Prior research on hallucination evaluation for both images (Li et al., 2023e; Wang et al., 2023; Rohrbach et al., 2018) and videos (Wang et al., 2024b; Yang et al., 2024; Guan et al., 2024) has predominantly focused on common visual aspects such as action- and object-based hallucinations. However, video-based hallucinations may involve additional dynamic factors associated with spatio-temporal patterns, which these studies overlook. In light of this, we propose to focus on the following five aspects to ensure comprehensive coverage of temporal concepts. Specifically, the first four aspects address hallucinations based on lexical semantics, while the fifth targets clause semantics.

- **Attribute (L-Sem)** describes the fine-grained characteristics of objects or subjects in the video. We additionally categorize this aspect into sub-aspects of *Size*, *Shape*, *Color*, *Count* and *State Change*.

- **Object (L-Sem)** relates to the interactions between objects and entities within the video. We further delineate this aspect into two fine-grained sub-aspects: *Object Recognition*, identifying the objects engaged in interactions, and *Interaction Classification* which concentrate on how these objects interact with other objects or subjects.

- **Action (L-Sem)** refers to the movements and behaviours exhibited by entities.

- **Direction (L-Sem)** indicates the orientation and movement trajectory of subjects or objects.

Table 1: Comparison of our benchmark dataset with existing video-based reasoning and hallucination evaluation datasets. For datasets with multiple evaluation tasks, only those relevant to hallucination evaluation are included. VL Entailment denotes the task of *video-language entailment*, while *Event Ordering* prompts the model to determine the chronological sequence of scenes in a video.

| | Dataset | Temporal Aspects | | | | | | | | | | Task Formats | Evaluation Metrics |
|---|---|---|---|---|---|---|---|---|---|---|---|---|---|
| | | Action | Attribute | | | | | Direction | Object | | Order | | |
| | | | Size | Shape | Color | Count | State-Change | | Recognition | Interaction | | | |
| *Video Reasoning* | SEEDBench (Li et al., 2023a) | ✓ | ✗ | ✗ | ✗ | ✗ | ✗ | ✗ | ✗ | ✗ | ✓ | MCQA | Accuracy |
| | VideoBench (Ning et al., 2023) | ✓ | ✓ | ✓ | ✓ | ✓ | ✗ | ✗ | ✓ | ✗ | ✗ | MCQA | Accuracy |
| | MVBench (Li et al., 2024b) | ✓ | ✗ | ✗ | ✗ | ✗ | ✗ | ✓ | ✓ | ✓ | ✓ | MCQA | Accuracy |
| | Video-MME (Fu et al., 2024) | ✓ | ✓ | ✓ | ✓ | ✓ | ✗ | ✗ | ✓ | ✗ | ✗ | MCQA | Accuracy |
| *Hallucination Evaluation* | Vript (Yang et al., 2024) | ✓ | ✗ | ✗ | ✗ | ✗ | ✗ | ✗ | ✓ | ✓ | ✓ | Video Captioning Event Ordering | F1 Score Accuracy |
| | VideoCon (Bansal et al., 2024) | ✓ | ✓ | ✓ | ✓ | ✓ | ✗ | ✗ | ✓ | ✗ | ✓ | VL Entailment | ROC-AUC |
| | HallusionBench (Guan et al., 2024) | ✓ | ✗ | ✗ | ✗ | ✗ | ✗ | ✓ | ✗ | ✗ | ✓ | Y/N QA | Accuracy |
| | VIDHAL (Ours) | ✓ | ✓ | ✓ | ✓ | ✓ | ✓ | ✓ | ✓ | ✓ | ✓ | MCQA, Caption Ordering | Accuracy NDCG |

- **Event Order (C-Sem)** represents the correct sequence of events in the video. During our collection, we retain videos that contain at least three distinct events.

The choice of these aspects is motivated by their collective corroboration across independent video-based studies (Wang et al., 2024b; Liu et al., 2024g; Li et al., 2024b), which identify them as critical dimensions for robust spatio-temporal reasoning in VLLMs. Crucially, while certain aspects are not necessarily specific to videos (*e.g.*, Object and Attribute), VIDHAL extends them beyond static spatial settings. For instance, attributes are evaluated on how they change dynamically over time rather than assessing them as fixed visual properties. We present an example that illustrates the direction aspect in Figure 2, with additional examples available in the supplementary material.

### 3.3 Hallucinatory Caption Generation

Based on the aspects defined in Section 3.2, we construct our benchmark from four public video understanding datasets: TempCompass (Liu et al., 2024g), Perception Test (Patraucean et al., 2023), MVBench (Li et al., 2024b), and AutoEval-Video (Chen et al., 2023). TempCompass and MVBench provide extensive coverage of all five temporal aspects, while Perception Test and AutoEval-Video focus on human-object interactions and attribute changes, respectively.

Existing hallucination benchmarks (Li et al., 2023e; Wang et al., 2023) rely mostly on binary questions for evaluation, limiting their efficacy in detecting subtle video hallucinations, such as minor event inconsistencies. To address this issue, we advocate a novel evaluation protocol incorporating several carefully annotated captions. Specifically, each video will be annotated with $M$ captions that reflect varying degrees of hallucination in VLLMs. Given the cost and labor intensity of manual annotation, we follow existing benchmark studies such as PhD (Liu et al., 2024e) and MVBench (Li et al., 2024b), opting for automatic caption generation using a carefully designed pipeline illustrated in Figure 2.

**Anchor Caption Generation.** The video instances in VIDHAL are sourced from various public datasets, resulting in distinct associated metadata such as long-form captions in AutoEval-Video and question-answer pairs in MVBench. To ensure structure consistency and information granularity in the respective dataset description across all instances, we automatically generate an anchor caption for each video. Specifically, we input the metadata for each video $V^i$ into GPT-4o and prompt it to generate a concise and accurate description $y_+^i$ using the provided metadata information.

**Hallucinatory Caption Generation.** After obtaining the positive caption for each video instance, we augment the dataset with $M-1$ additional captions containing hallucinated content. For a given video instance $V^i$, we construct a set $\mathcal{Y}_-^i = \{y_-^{i,1}, \cdots, y_-^{i,M-1}\}$ containing captions with different levels of hallucination based on the temporal concepts associated with it. Specifically, $y_-^{i,k}$ exhibits heavier hallucination than $y_-^{i,j}$ for caption hallucination degree $j < k$. We leverage GPT-4o to generate $\mathcal{Y}_-^i$ by combining the anchor caption

Figure 3: Human agreement on hallucination levels in the VIDHAL dataset. (Left) Distribution of agreement ratios per video sample. (Right) Average agreement ratio for each aspect, with an overall average of 87%. The *"Complete Agreement"* dotted line at 1.0 denotes the upper bound where the constructed caption orderings are consistent with human consensus rankings across all validated instances.

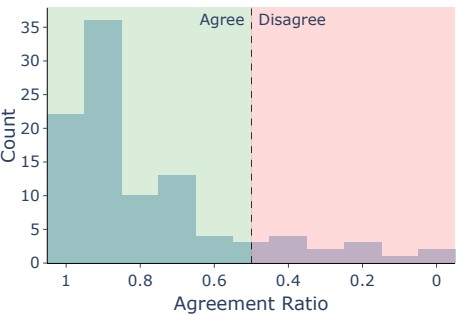
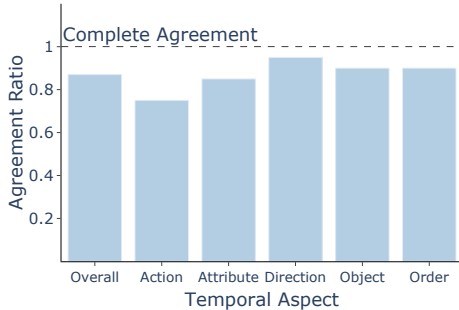

$y^i_+$ and prompting it to create $y^{i,1}_-, \cdots, y^{i,M-1}_-$ progressively in increasing levels of hallucination, where the prompt includes detailed guidelines and definitions for the different grades of hallucination severity. Given the distinct characteristics of each aspect, hallucination severity is operationalized in terms of the likelihood of confusing the ground-truth aspect value with a plausible alternative, with aspect-specific criteria detailed in Table 4 in the Appendix. To steer GPT-4o towards generating captions that faithfully reflect the intended hallucination severity levels, the prompt further includes aspect-specific in-context examples that adhere to these operationalized criteria. The set of captions associated with $V^i$ is then defined as $\mathcal{Y}^i \leftarrow \{y^i_+\} \bigcup \mathcal{Y}^i_-$ consisting of both the anchor and hallucinatory captions. The full anchor and hallucinatory caption generation prompts, along with aspect-specific in-context examples, are detailed in Figures 15–21 in the Appendix.

### 3.4 Dataset Statistics and Human Validation

Using our automatic annotation pipeline, our VIDHAL benchmark consists of a total of 1,000 video instances each tagged with $M = 3$ captions. As shown in Table 1, our VIDHAL dataset stands out from other video understanding (Li et al., 2023a; Ning et al., 2023; Li et al., 2024b; Fu et al., 2024) and hallucination benchmarks (Guan et al., 2024; Liu & Wan, 2023) in terms of two dimensions: I) VIDHAL encompasses a diverse range of video-centric temporal aspects; and II) We introduce a novel caption ordering task along with two tailored metrics to capture subtle hallucinations previously ignored by paired questions.

To validate the reliability of our generated captions, we randomly sampled 100 examples for human evaluation, with each example assessed by an average of 15 annotators. The validation process focused on verifying whether the ranking of hallucinatory captions produced by our pipeline aligns with human judgment. Specifically, the Inter-Rater agreement rate is computed as the proportion of validated instances in which our automatically generated caption ordering matches the consensus ranking derived from human annotations, where the consensus is determined by a majority vote among individual annotator rankings. As shown in Figure 3, an overall agreement rate of 87% was achieved, demonstrating strong consistency with human preferences across all temporal aspects.

## 4 VidHal Evaluation Protocol

To address the limitations of binary question-based benchmarks, we propose two evaluation tasks: *multiple-choice question answering* and a novel *caption ordering* task, detailed in Section 4.1. Additionally, we develop corresponding metrics to comprehensively measure hallucinations in video-based VLLMs in Section 4.2.

### 4.1 Evaluation Tasks

**Multiple-Choice Question Answering (MCQA)** assesses the model's spatiotemporal understanding in a coarse-grained manner. Specifically, the VLLM is provided with a video $V^i$ and its corresponding set of

captions $\mathcal{Y}^i$ as answer options and instructed to select the most appropriate caption for the video.

**Caption Ordering** evaluates a model's visual reasoning from a nuanced granularity, instructing VLLMs to order the provided captions based on their hallucination level. Through pairwise comparisons across all captions, this task identifies cases where the model struggles to distinguish varying levels of hallucination severity beyond anchor-hallucination distinctions.

Specifically, we design two caption ordering sub-tasks. The first, *naive caption ordering*, requires VLLMs to rank all captions at once. However, this sub-task can confuse several VLLMs due to its inherently challenging nature and the inferior instruction-following capabilities of some models. As a complement, we propose an additional sub-task, *relative caption ordering*, which decomposes the prior task into multiple paired caption ordering tasks. Since each paired ordering task is answered in isolation, the VLLM may produce a non-transitive, cyclic ranking.

To circumvent this, we query the model with consecutive caption pairs, prompting the final pair only if multiple orderings are possible. For instance, given captions $A$, $B$, and $C$, if the model predicts $A \prec B$ and $B \prec C$, the overall order $A \prec B \prec C$ can be directly inferred. However, if it instead ranks $B \prec A$ , as shown in Figure 4, we additionally include a third comparison between $A$ and $C$ to resolve any ambiguity in determining in the final order. Notably, our relative caption ordering task is more challenging than previous binary questions. This complexity

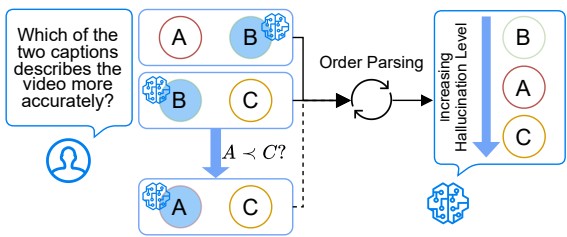

Figure 4: Visual illustration of the *relative caption ordering* task in VIDHAL.

arises from certain paired questions in VIDHAL where both options are hallucinatory, making them harder to distinguish as opposed to ⟨*positive, hallucinatory*⟩ pairs.

## 4.2 Evaluation Metrics

**Notations** For a particular video instance $V^i$, we define the ground truth caption order for $V^i$ to be $\mathcal{Y}^i_* = (y^i_+, y^{i,1}_-, \cdots, y^{i,M-1}_-)$. Further let the $j^{th}$ element in this ordering be indexed as $\mathcal{Y}^{i,j}_*$.

**MCQA** We employ the standard accuracy metric for the MCQA task:

$$\text{Accuracy} = \frac{1}{N} \sum_{i=1}^{N} \mathbb{I}\left[ R_{MCQA}(V^i, \mathcal{Y}^i) = y^i_+ \right], \tag{1}$$

where $N$ is the number of video instances, $\mathbb{I}$ denotes the indicator function, and $R_{MCQA}(V^i, \mathcal{Y}^i)$ represents the best matched caption from $\mathcal{Y}^i$ for $V^i$ as predicted by a VLLM.

**Caption Ranking** Inspired by metrics from the information retrieval domain (Gao et al., 2023), we adapt the well-established Normalized Discounted Cumulative Gain (NDCG) (Järvelin & Kekäläinen, 2002) for hallucination assessment in VIDHAL. Unlike previous metrics like POPE (Li et al., 2023e), our metric awards partial credit for correctly ordered caption pairs even when the optimal ranking is not achieved. As such, we expect the metric to effectively capture and distinguish both subtle and severe hallucinations generated by video-based VLLMs. Formally, we define our adapted NDCG metric as follows:

$$\text{NDCG} = \frac{1}{N} \sum_{i=1}^{N} \frac{\text{DCG}_i - \text{rDCG}_i}{\text{iDCG}_i - \text{rDCG}_i}, \tag{2}$$

where $\text{DCG}_i$ is formulated as:

$$\text{DCG}_i = \sum_{j=1}^{M} \frac{r\left(\hat{y}^{i,j}, \mathcal{Y}^i_*\right)}{\log(j+1)}, \tag{3}$$

and $\hat{y}^{i,j}$ represents $j^{th}$ caption in the ranked order predicted by the VLLM. The perfect ordering is achieved when $\hat{y}^{i,1} = y_+^i$ and $\{\hat{y}^{i,j} = y_-^{i,j-1}\}_{j=2 \to M}$. To evaluate predicted caption orders relative to this ideal sequence, a relevance function $r\left(\hat{y}^{i,j}, \mathcal{Y}_*^i\right)$ is designed to assign higher scores to $\hat{y}^{i,j}$ with lower hallucinatory extent:

$$r(\hat{y}^{i,j}, \mathcal{Y}_*^i) = M + 1 - \text{pos}(\hat{y}^{i,j}, \mathcal{Y}_*^i), \tag{4}$$

where $\text{pos}(\hat{y}^{i,j}, \mathcal{Y}_*^i)$ denotes the position of $\hat{y}^{i,j}$ in $\mathcal{Y}_*^i$. Finally, $\text{DCG}_i$ is normalized to a range of $[0, 1]$ using $\text{iDCG}_i$ and $\text{rDCG}_i$, with a score of 1 indicating perfect alignment of the predicted order with $\mathcal{Y}_*^i$. Specifically, these terms represent the maximum and minimum $\text{DCG}_i$ scores obtained from the optimal ordering $\mathcal{Y}_*^i$ and its reverse, respectively,

$$\text{iDCG}_i = \sum_{j=1}^{M} \frac{r\left(\mathcal{Y}_*^{i,j}, \mathcal{Y}_*^i\right)}{\log(j+1)}, \ \text{rDCG}_i = \sum_{j=1}^{M} \frac{r\left(\mathcal{Y}_*^{i,M-j}, \mathcal{Y}_*^i\right)}{\log(j+1)}. \tag{5}$$

## 5 Experiments

### 5.1 Experimental Settings

**Models.** We evaluated twenty-three VLLMs from thirteen different model families, including ten open-source models: VideoChat (Li et al., 2023d), LLaMA-VID (Li et al., 2024c), VideoChat2 (Li et al., 2024b), mPLUG-Owl3 (Ye et al., 2024), LLaVA-NeXT-Video (Zhang et al., 2024), VideoLLaMA2 (Cheng et al., 2024), MiniCPM-V (Yao et al., 2024), LongVU (Shen et al., 2024), InternVL2.5 (Chen et al., 2024c) and Qwen2.5-VL (Bai et al., 2025), and three proprietary models: GPT-4o (OpenAI, 2023), GPT-4.1 and Gemini (Reid et al., 2024; Comanici et al., 2025). These models represent a wide variety of architectural designs and training paradigms. Additionally, we included a random baseline that selects and ranks candidate options randomly.

**Implementation Details.** All experiments were conducted using four NVIDIA A100 40GB GPUs and inference APIs. The input captions in $\mathcal{Y}^i$ were randomized using a fixed, predefined randomization seed across experiments. We adhered to the inference and model hyperparameters outlined in the respective original models, and employed greedy decoding during generation for a fair comparison.

### 5.2 Overall Results

**Benchmark Results.** We present the overall results of representative VLLMs in Table 2 across both MCQA and caption ordering tasks. We make three key observations from this table:

*Competitive Performance of Open-Source Models.* Open-source VLLMs achieve performance comparable to proprietary models, particularly on MCQA and relative caption ordering tasks. Notably, LongVU achieves the highest performance among open-source models and surpasses strong proprietary models such as GPT-4o, GPT-4.1, and Gemini-1.5 on these tasks.

*Parameter Scale vs. Performance.* Among open-source VLLMs, smaller variants (e.g., 7B parameter models) outperform their larger counterparts within the same model family, as observed with InternVL2.5 and Qwen2.5-VL. This suggests that simply increasing model capacity may provide limited benefits for reducing video-based hallucinations in current VLLM development. While LLaVA-NeXT-Video and VideoLLaMA2 use different backbone LLMs in their larger variants, the performance gains observed are more likely attributable to increased parameter scale rather than the choice of LLM.

*Impact of Architecture Design.* Model families that achieve high scores across both tasks often incorporate design efforts specifically targeting visual understanding, such as dynamic resolution scaling (InternVL2.5, Qwen2.5-VL) and temporal reduction techniques (LongVU). These findings may suggest that specialized architectural innovations are key factors in mitigating temporal hallucinations.

**Aspect-aware Results.** Figure 5 (Left) highlights the fine-grained, aspect-specific performance of the notable VLLMs. Notably, VLLMs demonstrate substantially stronger results on the *Action* and *Object*

Table 2: Overall benchmark performance of VLLMs on our VIDHAL dataset. #Params refers to the number of parameters of the base LLM used. The best performance for each task is highlighted in **bold** for open-source models, and underlined for proprietary models.

| Model | Vision Encoder | LLM | #Params | #Frames | Accuracy | NDCG Naive | NDCG Relative |
|---|---|---|---|---|---|---|---|
| *Baseline* | | | | | | | |
| Random | - | - | - | - | 0.326 | 0.505 | 0.480 |
| *Open-Source Models* | | | | | | | |
| VideoChat | EVA-CLIP-G | Vicuna | 7B | 8 | 0.381 | 0.475 | 0.488 |
| LLaMA-VID | EVA-CLIP-G | Vicuna | 7B | 1fps | 0.358 | 0.486 | 0.521 |
| VideoChat2 (Vicuna) | UMT-L | Vicuna | 7B | 16 | 0.426 | 0.486 | 0.577 |
| VideoChat2 (Mistral) | UMT-L | Mistral | 7B | 16 | 0.443 | 0.503 | 0.475 |
| VideoChat2 (Phi) | UMT-L | Phi3 | 3.8B | 16 | 0.514 | 0.626 | 0.612 |
| mPLUG-Owl3 | SigLIP/SO400M | Qwen2 | 7B | 16 | 0.596 | 0.641 | 0.707 |
| LLaVA-NeXT-Video (7B) | SigLIP/SO400M | Vicuna | 7B | 32 | 0.509 | 0.518 | 0.620 |
| LLaVA-NeXT-Video (32B) | SigLIP/SO400M | Qwen1.5 | 32B | 32 | 0.663 | 0.641 | 0.747 |
| VideoLLaMA2 (7B) | CLIP ViT-L/14 | Mistral | 7B | 8 | 0.541 | 0.564 | 0.622 |
| VideoLLaMA2 (72B) | CLIP ViT-L/14 | Qwen2 | 72B | 8 | 0.647 | 0.787 | 0.760 |
| MiniCPM-V 2.6 | SigLIP/SO400M | Qwen2 | 7B | 1fps | 0.377 | 0.530 | 0.523 |
| LongVU | SigLIP/SO400M | Qwen2 | 7B | 1fps | **0.795** | 0.453 | **0.846** |
| InternVL2.5 (8B) | InternViT-300M (V2.5) | InternLM2.5 | 7B | 16 | 0.773 | 0.475 | 0.827 |
| InternVL2.5 (26B) | InternViT-6B (V2.5) | InternLM2.5 | 20B | 16 | 0.742 | 0.498 | 0.775 |
| Qwen2.5-VL (7B) | Qwen2.5-ViT | Qwen2.5 | 7B | 1fps | 0.76 | **0.825** | 0.826 |
| Qwen2.5-VL (32B) | Qwen2.5-ViT | Qwen2.5 | 32B | 1fps | 0.732 | 0.811 | 0.800 |
| Qwen2.5-VL (72B) | Qwen2.5-ViT | Qwen2.5 | 72B | 1fps | 0.74 | 0.807 | 0.793 |
| *Proprietary Models* | | | | | | | |
| GPT-4o | - | - | - | 1fps | 0.772 | 0.840 | 0.826 |
| GPT-4.1 | - | - | - | 1fps | 0.777 | 0.845 | 0.834 |
| Gemini-1.5 (Flash) | - | - | - | 1fps | 0.657 | 0.738 | 0.745 |
| Gemini-1.5 (Pro) | - | - | - | 1fps | 0.671 | 0.765 | 0.753 |
| Gemini-2.5 (Flash) | - | - | - | 1fps | 0.814 | 0.875 | 0.860 |
| Gemini-2.5 (Pro) | - | - | - | 1fps | 0.814 | 0.876 | 0.861 |

Figure 5: Performance of VLLMs across individual aspects and sub-aspects in VIDHAL. (Left) Aspect-specific NDCG scores for the naive and relative caption ordering. (Middle) NDCG scores for *Attribute*, and (Right) *Object* sub-aspects in relative caption ordering.

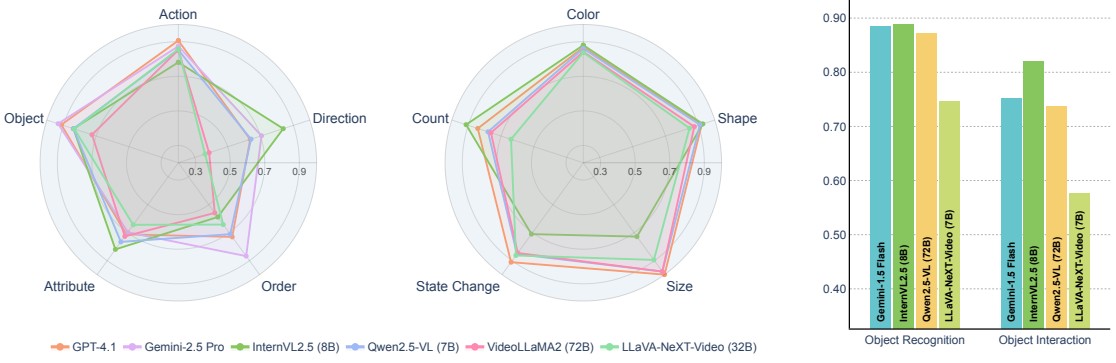

aspects compared to others. This can likely be attributed to current visual instruction tuning datasets predominantly emphasizing object-centric recognition and coarse-grained activity classification, potentially encouraging strong reliance on image-based priors when generating predictions. In contrast, these models tend to underperform on temporally nuanced aspects such as direction and event order, which are inherently unique to the video modality.

We further analyzed the distribution of results for the relative caption ranking task across sub-aspects of the *Attribute* and *Object* aspects in the middle and rightmost subfigures in Figure 5. While VLLMs

generally maintain consistent performance across *Attribute* sub-aspects, their effectiveness declines slightly when reasoning about *Count* and *Color*, suggesting that reasoning over such fine-grained visual properties remains challenging for VLLMs. For the *Object* aspect, several models performed significantly worse in *Interaction Classification* than in *Object Recognition*, highlighting the need to better model object interactions to bridge the gap between recognition and understanding.

### 5.3 Ablation Studies

We conducted further experiments to investigate VLLM behaviour beyond the aggregate benchmark results presented in Section 2, analysing model behaviour and failure modes in greater detail. Specifically, we begin by characterising *where* models fail, examining pairwise error rates across caption pairs of varying hallucination levels to identify where nuanced discrimination breaks down. Next, we probe a plausible underlying reason for *why* such failures arise, exploring the extent to which models over-rely on single-frame spatial information rather than leveraging full temporal context. Finally, we validate the robustness of our evaluation protocol against potential artifacts introduced by the relative caption ordering task.

**Hallucination Differentiation Sensitivity.** To characterise *where* models fail, we studied VLLM robustness in differentiating caption pairs of varying hallucination levels, beyond overall ordering performance. Specifically, we identify how the tendency of VLLMs to favour captions with higher hallucination over those with lower degree varies as the difference in hallucination severity between the compared captions changes. Formally, for two captions with different hallucination levels $j, k$ where $j > k$, we introduce the following metric to quantify such *hallucination misalignment* cases:

$$HM_{j \to k} = \frac{1}{N} \sum_{i=1}^{N} \mathbb{I}\left[\mathcal{Y}_*^{i,j} \prec \mathcal{Y}_*^{i,k}\right], \tag{6}$$

which reflects the proportion of cases in which the VLLM selects the caption with a higher level of hallucination $j$ over $k$. Specifically, we examine three key cases: when the most hallucinatory caption is chosen over both the lower-hallucination and anchor captions, and when the lower-hallucination caption is selected over the anchor caption. These cases are represented by $HM_{3 \to 1}$, $HM_{3 \to 2}$, and $HM_{2 \to 1}$, respectively, with results presented in Figure 6.

Our findings show that advanced VLLMs, such as VideoLLaMA2 (72B), GPT-4.1 and Qwen2.5-VL models can generally distinguish

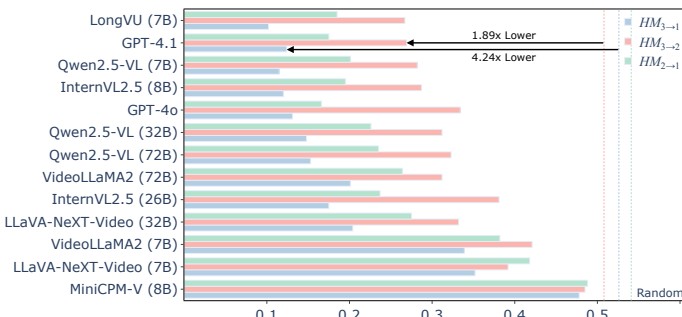

Figure 6: Hallucination misalignment (HM) scores on VIDHAL, with *Random* representing HM scores from the random baseline.

positive captions from severely hallucinated ones, reflected by their low $HM_{3 \to 1}$ scores in Figure 6. However, two key observations emerge from our experiments: First, most VLLMs struggle to differentiate the lower hallucinatory caption from the anchor, as evidenced by the gap between $HM_{3 \to 1}$ and $HM_{2 \to 1}$. Second, all models exhibit high $HM_{3 \to 2}$ scores, indicating difficulty in distinguishing between two hallucinatory captions with varying degrees. These results suggest gaps in nuanced video reasoning may contribute to hallucinatory behavior in VLLMs, a challenge not addressed by existing ⟨*positive, hallucinatory*⟩-based evaluation methods. (Li et al., 2023e; Wang et al., 2024b; Guan et al., 2024).

**Image Prior Reliance.** Next, we investigated a plausible underlying cause for *why* such deficiencies are observed, examining whether poor performance stems from models over-prioritising spatial over temporal information. Previous research shows that VLLMs often rely on image priors for reasoning (Lei et al., 2023; Buch et al., 2022), overlooking key spatiotemporal features. This is exemplified by dominant influence of a few frames on response generation. To examine how this bias affects video-based hallucinations, we used a video summarization algorithm (Son et al., 2024) to extract the most salient frame $v^i$ from $V^i$. We then generated VLLM responses on VIDHAL using $v^i$ instead of $V^i$ as visual input. The effect of image priors

Figure 7: Overlapping ratios of model predictions under single-frame and full-video inputs for correct, incorrect and overall predictions in the (Left) naive and (Right) relative caption ordering tasks. *Complete Reliance* indicates that the VLLM always produces the same response for both video and single frames.

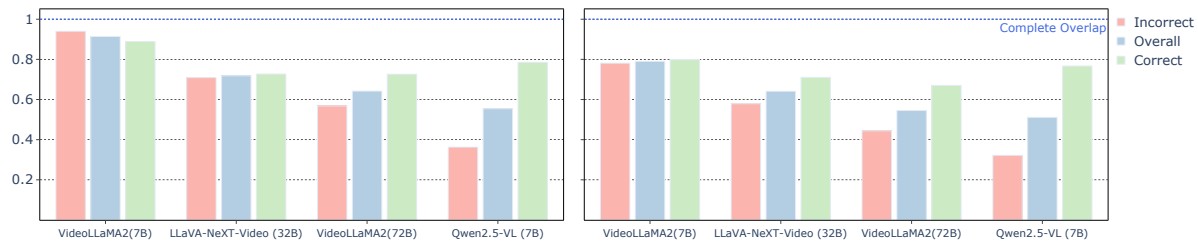

Table 3: Proportion of VIDHAL instances where the predicted caption ordering $\hat{\mathcal{Y}}$ is cyclic.

| Metric | VideoChat2 (7B) | mPLUG-Owl3 | LLaVA-NeXT-Video (7B) | VideoLLaMA2 (7B) |
|---|---|---|---|---|
| $P(\text{IsCyclic}(\hat{\mathcal{Y}}))$ | 0.073 | 0.030 | 0.037 | 0.040 |

is evaluated by identifying overlapping instances where responses from $V^i$ and $v^i$ remain consistent across both correct and incorrect orderings. As shown in Figure 7, results reveal that VLLMs heavily rely on image priors, particularly in smaller models where overlap ratios are consistently high across correct, incorrect, and overall predictions alike. With increasing model capacity, the overlap ratios for incorrect predictions decrease, suggesting that larger models make better use of additional temporal information to correct errors, reflecting more robust spatio-temporal reasoning. Notably, Qwen2.5-VL (7B) demonstrates comparatively lower overlap ratios, particularly for incorrect predictions, relative to models of similar or even greater parameter scale, indicating a stronger capacity to leverage full-video temporal cues to refine its predictions.

**Transitive Robustness.** Lastly, since the relative caption ordering task decomposes ordering into sequential pairwise comparisons, we verify that cyclic preferences do not confound the evaluation scores. Although the relative caption ranking task enhances VLLM stability in caption ordering, it may reveal transitive inconsistencies between the final caption order and individual paired orders. For instance, the predicted order $\hat{\mathcal{Y}}^i = [\hat{y}^{i,1}, \hat{y}^{i,2}, \hat{y}^{i,3}]$ may be inferred when the VLLM predicts $\hat{y}^{i,1} \prec \hat{y}^{i,2}$ and $\hat{y}^{i,2} \prec \hat{y}^{i,3}$, but it may predict $\hat{y}^{i,3} \prec \hat{y}^{i,1}$ when prompted to order these two captions, contradicting the initial order. We quantify such errors by measuring the proportion of instances where $\hat{\mathcal{Y}}^i$ fails to entail $\hat{y}^{i,3} \prec \hat{y}^{i,1}$. This is defined as:

$$P(\text{IsCyclic}(\hat{\mathcal{Y}})) = \frac{1}{N} \sum_{i=1}^{N} \mathbb{I}\left[ R_{MCQA}(V^i, \{\hat{y}^{i,1}, \hat{y}^{i,3}\}) = \hat{y}^{i,3} \right]. \tag{7}$$

The results, presented in Table 3, demonstrate that even the least performant models exhibit strong consistency between pairwise caption preferences and final caption orderings. These results indicate that our proposed caption-ordering task is robust against anomalous scoring stemming from unintended cyclic preferences.

## 5.4 Qualitative Results

We conducted a qualitative analysis of VLLM-generated responses for the caption ordering task to complement the quantitative results with deeper insight into model behaviour across task formats. We randomly sampled representative instances across the temporal aspects, evaluated selected VLLMs on both the naive and relative caption ordering tasks, and recorded their predicted orderings for direct comparison within each instance. The Action, Attribute, and Order aspects were selected for illustration in Figure 8. For each instance, the three captions are labelled A, B, and C in increasing order of hallucination level, with the ground truth ordering defined as $A \succ B \succ C$. We assess reasoning strength and consistency by examining the number of adjacent swaps required to transform each predicted ordering into the optimal one. Our analysis reveals two notable patterns:

*Relative caption ordering consistently elicits more accurate VLLM responses compared to naive ordering*, with this improvement being more pronounced in weaker models. The enhanced performance is evidenced by increased ordering correctness when transitioning from naive to relative ordering, as pairwise comparison prompts enable models to make finer-grained distinctions between caption quality levels.

*Advanced VLLMs demonstrate greater stability across both ordering paradigms*, exhibiting lower variance in predictions between both ordering tasks. Stronger models such as VideoLLaMA2 (72B) and GPT-4o generally require at most one swap to reach the optimal ordering, consistent with their higher NDCG scores in Table 2. Furthermore, their predicted orderings under naive and relative caption ordering differ by at most one positional swap, as observed in the Attribute and Action aspects. This consistency suggests that stronger models maintain more coherent internal representations of caption quality across varying task formats.

## 6 Conclusion

**Summary.** In this work, we introduce the VIDHAL benchmark to address gaps in the video-based hallucination evaluation of VLLMs. VIDHAL features video instances spanning five temporal aspects. Additionally, we propose a novel caption ordering evaluation task to probe the fine-grained video understanding capabilities of VLLMs. We conduct extensive experiments on VIDHAL through the evaluation of twenty-three VLLMs, exposing their limitations in unexpected hallucination generation. Our empirical results shed light on several promising directions for future work: *e.g.*, incorporating a broader range of temporal features during pretraining and mitigating single-frame priors to enhance temporal reasoning. These advancements will help to address the hallucination problem in video-based VLLMs, enhancing their robustness for real-world video understanding applications.

**Limitations.** We acknowledge that the VIDHAL evaluation suite relies on synthetic captions generated by GPT-4o, which may contain biases inherently present in the model. We note that this design choice is consistent with prior research, as several established language-only and vision-language benchmarks similarly use GPT-4o for dataset construction (Liu et al., 2024e; Li et al., 2024a;b; 2023a;c) or response evaluation (Guan et al., 2024; Sun et al., 2024; Liu et al., 2024a). To reduce over-alignment to GPT-4o's preferences, we incorporate additional strong LLMs, including Gemini-1.5 (Reid et al., 2024) and LLaMA2 (70B) (Touvron et al., 2023) to assess and filter generated captions. We further conduct a final step of manual verification and editing to address residual misalignments not captured by automated filtering. Full details of this post-processing pipeline are provided in Appendix A.2. While these measures enhance annotation robustness, fully eliminating LLM-induced biases in synthetic caption generation remains an open challenge. Moreover, while VIDHAL is designed to assess both subtle and severe hallucinations across a wide range of temporal aspects, the generated hallucinatory captions may not fully capture the internal tendencies of all evaluated VLLMs, a caveat researchers should bear in mind when interpreting results.

VIDHAL assesses a model's ability to identify hallucinated content in externally provided captions, rather than directly measuring hallucinations in model-generated outputs, and we acknowledge this inherent gap

Figure 8: Qualitative examples of VLLM responses on the caption ordering tasks, for the *Attribute*, *Order* and *Action* aspects. Captions are labelled *A*, *B*, and *C* in increasing order of hallucination level, with the ground truth ordering $A \succ B \succ C$. Predicted orderings are shown for both naive and relative caption ordering across representative VLLMs.

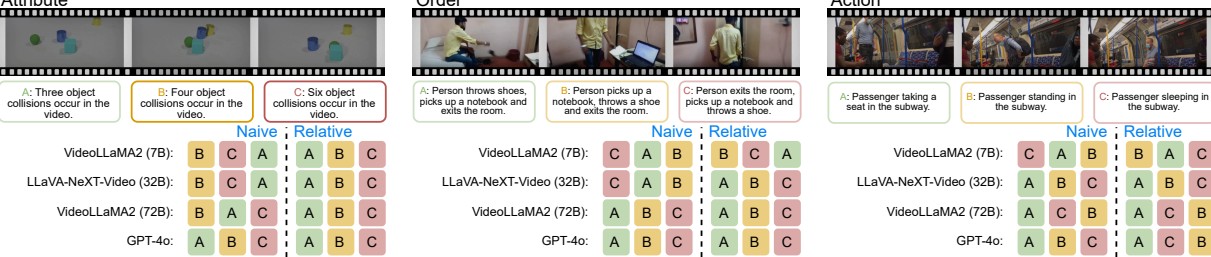

between discriminative evaluation and generative hallucination. We note that this evaluation paradigm follows from prior work, with established benchmarks (Li et al., 2023e; Wang et al., 2023; 2024b; Li et al., 2024a) similarly adopting discriminative approaches for their capacity for controlled and stable assessment. Crucially, existing hallucination mitigation methods (Huang et al., 2024; Zhou et al., 2024c;d) evaluated on both discriminative benchmarks and open-ended generative tasks consistently show that improvements on the former correlate with reductions in generative hallucinations, supporting discriminative evaluation as a valid proxy for assessing hallucination generation tendencies in VLLMs.

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

# Appendix

## A Benchmark Construction Details

### A.1 Dataset Statistics

Figure 9: Distribution of visual instances in VIDHAL by (Left) public dataset source, categorized by the five temporal aspects, and (Right) temporal aspects and their sub-aspects.

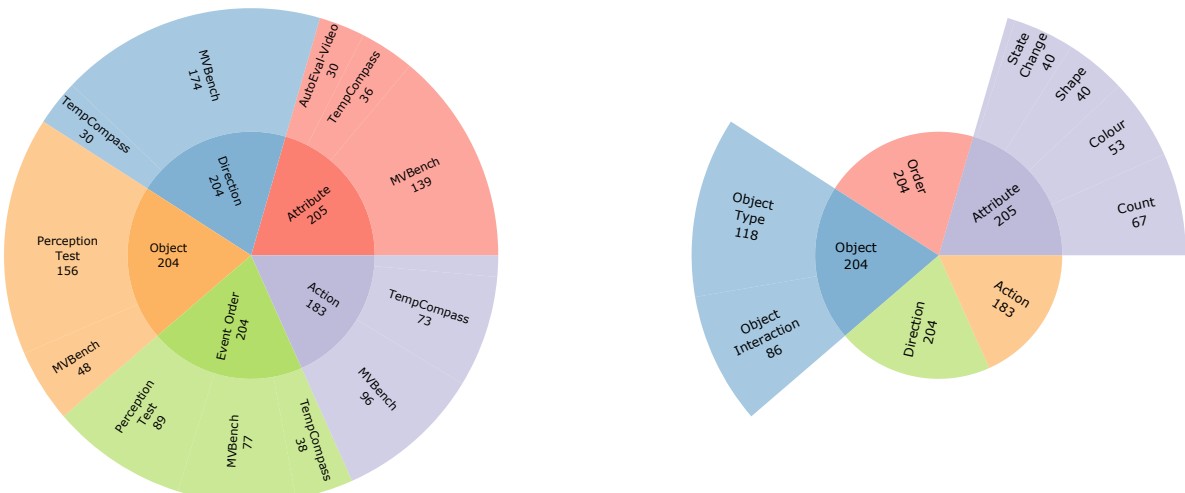

Figure 9 presents the distribution of visual instances in VIDHAL by public dataset sources and temporal aspects. Additionally, Figure 10 further shows the distribution of ground truth answers for the MCQA and caption ordering tasks. One can observe that both temporal aspects and ground truth options are uniformly distributed across our benchmark. The distribution of video caption lengths and video durations is also presented in Figure 11.

### A.2 Dataset Development Pipeline

**Visual Instance Selection** To ensure a rich coverage of temporal aspects and visual diversity, we methodically selected video instances from four public datasets: TempCompass Liu et al. (2024g), Perception Test Patraucean et al. (2023), MVBench Li et al. (2024b), and AutoEval Video Chen et al. (2023). Given the unique characteristics of each dataset, we outline the specific guidelines adopted for each dataset below:

Figure 10: Distribution of (Left) correct answer options for the MCQA task and (Right) optimal option orders for the caption ordering task.

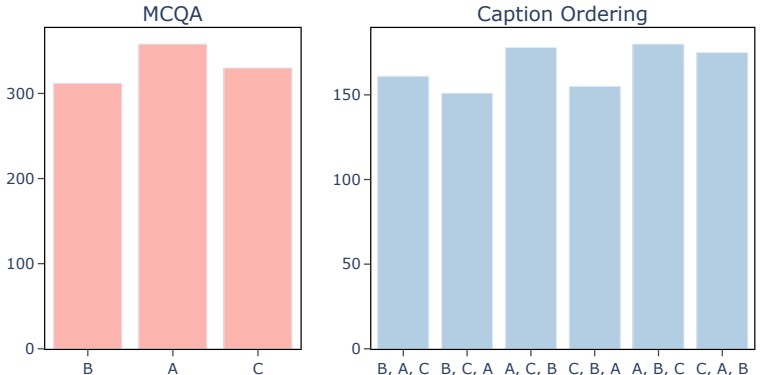

Figure 11: Distribution of (Left) caption lengths with an average of 11.2 words, and (Right) duration of videos in VIDHAL with an average of 15.8s.

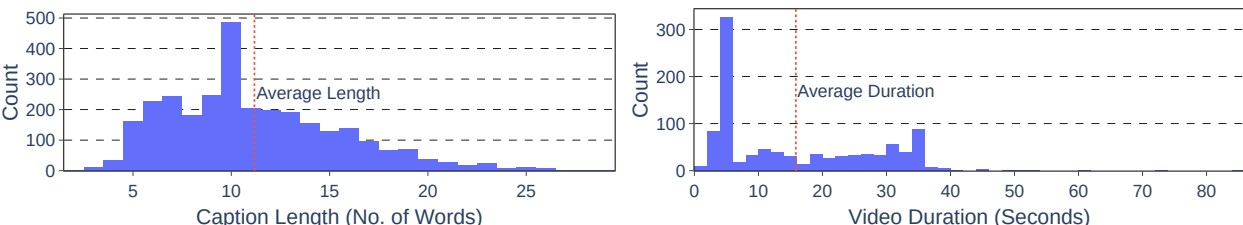

- **TempCompass** encompasses five temporal aspects: *Action*, *Speed*, *Direction*, *Event Order*, and *Attribute Change*. As most of these aspects align with those chosen to construct VIDHAL, we retain all video instances except those related to speed. TempCompass includes four evaluation tasks: *MCQA*, *Yes/No QA*, *caption matching*, and *caption generation*. Given the conciseness of captions in the latter two tasks, their information can often be subsumed within the more detailed QA-based annotations. Therefore, we focus exclusively on MCQA and Yes/No QA annotations to create an informative anchor caption.

- **Perception Test** spans various skill and reasoning domains to thoroughly evaluate VLLMs' perception and understanding abilities. Our inspection of these evaluation dimensions reveals alignment between the *semantics*, *physics*, and *memory* skill areas, as well as *descriptive* and *explanatory* reasoning dimensions, with the temporal aspects of action, order, and event order. Accordingly, we limit our video selection in Perception Test to these specific pillars. Additionally, we review the question templates adopted in these areas and select video instances with question-answer pairs that support VIDHAL's evaluation objectives. The specific skills and associated questions chosen are detailed in Figure 12.

- **MVBench** includes twenty video understanding tasks with question-answer pairs designed to challenge the reasoning capabilities of VLLMs. Similar to the Perception Test, we identify the tasks relevant to the temporal aspects in VIDHAL and focus on collecting videos belonging from these tasks. The specific tasks for each aspect are presented in Figure13. We observe that MVBench contains repeated use of certain scenarios across tasks, indicated by similar question templates. To enhance caption diversity and minimize redundancy, we limit the number of examples for each unique scenario. The collected instances cover all five temporal aspects of VIDHAL.

- **AutoEval-Video** evaluates open-ended response generation in VLLMs through questions with detailed answers across nine skill dimensions. We focus on instances related to the *state transition* area, specifically assessing changes in object and entity attributes. For each instance, we retain the only answers to associated questions as they act as informative, long-form captions for the video.

Figure 12: Specific skills and corresponding questions from the Perception Test dataset chosen for VIDHAL instance selection, with the matched aspects indicated in brackets.

**Object Recognition [Object]:**
What object does the person use to hit other objects?
What ingredients did the person put in the bowl or on the plate?
Which object was removed by the person from the tabletop?
What geometric shapes did the person put on the table?
What objects did the person hit?
What is the order of the letters on the table at the end?
What letters did the person type on the computer in order?
**Distractor Action [Action]:**
What is the person preparing?
**Motion [Action]:**
What happens with the object after being placed on the slanted plane?
What happened once the person removed an object from the tabletop?

**Action Recognition [Action]:**
What object does the person use to hit other objects?
What objects did the person hit?
What is the person preparing?
Which statement describes better the actions done by the person?
**Sequencing [Event Order]:**
What letters did the person show in order?
What is the order of the letters at the end?",
In what order did the person put the objects in the backpack?
What is the order of the letters on the table at the end?

Figure 13: Evaluation tasks in MVBench aligned with temporal aspects in VIDHAL, categorized by aspect.

> **Action**: Action Sequence, Fine-Grained Action and Fine-Grained Pose
> **Direction**: Moving Direction.
> **Object**: Object Interaction, Object Existence.
> **Attribute**: Moving Attribute, Moving Count.
> **Order**: Action Sequence

**Incorrect Anchor Captions**  A minority of videos contain anchor captions misaligned with their content, often due to noisy metadata. Such discrepancies subsequently lead to undesirable hallucinatory captions. To remove such instances, we use BLIP2 Li et al. (2023b) to calculate frame-text matching scores across all video frames, selecting the maximum score as the representative video-text alignment score. Examples with incorrect anchor captions typically achieve low alignment scores, which are discarded as noisy instances.

**LLM-based Caption Generation**  We utilize GPT-4o's OpenAI (2023) text processing and generation capabilities to generate an anchor caption for each selected video, based on metadata from its original public dataset source. This metadata includes QA-based annotations for TempCompass, Perception Test, and MVBench, along with long-form answers for AutoEval-Video. The anchor caption is subsequently used as input for GPT-4o to generate corresponding hallucinatory captions.

To ensure the generated hallucinatory captions meet high-quality standards, we employ a detailed prompt adopting the following strategies to guide GPT-4o's output:

- Aspect-specific definitions which outline the characteristics of each aspect to be varied, prompting GPT-4o to modify anchor captions accordingly.

- Caption construction guidelines that define the structure, format, and hallucination levels required for the generated captions.

- In-context examples to illustrate the desired form of each hallucinatory caption for each aspect.

The prompts for generating anchor and hallucinatory captions are shown in Figures 14 to 17a, respectively, with definitions for each aspect are provided in Figure 16. Aspect-specific in-context examples are detailed in Figures 17b to 21. Separate in-context examples are provided for each *Attribute* subaspect of *Shape*, *Size*, *Color*, *Count*, and *State Change* to account for their distinct natures.

**Caption Filtering and Verification**  While we adopt carefully designed prompts with detailed severity guidelines and extensive aspect-specific in-context examples to control the generation of hallucinatory captions, GPT-4o may still introduce generative preferences that skew the hallucination ordering or reduce caption quality. To ensure the generated captions meet high quality standards and present reliable hallucination

Figure 14: Prompts used for generating the anchor caption from long-form captions.

> You are given a long caption describing the content of a video. Your task is to provide a summarised and concise version of this caption. Ensure that you keep all essential detail in the original caption.
>
> `<metadata>`
>
> Video description:

Figure 15: Prompt for generating aspect-specific hallucinatory captions based on anchor captions and in-context examples.

You are a chatbot tasked with generating hallucinatory captions for a video given the input ground truth caption provided. Your objective is to modify the `<aspect>` present in the provided caption to generate 2 incorrect captions of different levels of hallucination. `<aspect_definition>`. The extent of hallucination of each caption is measured on a scale of 1 to 3 in increasing levels of hallucination, with 1 denoting no hallucinations present and 3 denoting a large extent of hallucination. A description of the extent of hallucination represented by each score is given as follows:

1. The caption contains no hallucination. The caption that representing this score is the ground truth caption.
2. The caption includes moderate hallucination, describing an event that is different from the ground truth, yet possible given the context of the video
3. The caption contains high hallucination, describing an event that is realistic, but typically unlikely to happen given context reflected by the original caption.

The generated hallucinated captions should follow the guidelines below.

Guidelines:
1. Focus only on modifying the temporal aspect provided in the instruction. Do not change any other temporal aspect associated with objects or subjects in the video.
2. Keep your modifications brief but coherent. Your generated captions should be of similar length to the original caption.
3. Ensure that your generated captions depict realistic and believable scenarios even as they deviate from the original context. For example, avoid creating fictitious scenarios such as "Person flying on a broomstick" and "Monkey painting a picture".
4. You may rephrase the provided caption to maintain consistent sentence structure across all captions. However, make sure the factual content of the ground truth caption remains unchanged.
5. Each generated hallucinatory caption should be of the form `<score>`:`<caption>`, `<score>` takes a value from the hallucination scale defined and `<caption>` represents your provided hallucinatory caption.
6. No two generated <caption> should share the same `<score>`, and each caption should take on a unique level of hallucination from 2 to 3.

Here are some examples of how hallucinatory captions are expected to be constructed.

`<in_context_examples>`

Now, generate hallucinatory captions for the following video description.

Original Caption:
`<anchor_caption>`
Hallucinated Captions:

orderings, we adopt a two-stage process of automated filtering followed by manual verification. Specifically, we follow the steps below.

1. **Evaluation Criteria.** For each generated caption set, quality is assessed according to three criteria. *Realism* evaluates whether the hallucinated scenarios described in each caption are plausible and depict realistic events, ensuring that captions do not contain implausible or fictitious content. *Ordering Quality* assesses whether the relative hallucination levels assigned to the captions are appropriate, verifying that captions at higher severity levels do indeed reflect a greater degree of hallucination than those at lower levels. *Relevance* checks that deviations from the anchor caption are confined

Figure 16: Definitions incorporated into the prompt for generating hallucinatory captions for each aspect, with separate definitions provided for each sub-aspect in the *Attribute* aspect.

**Action**: Actions refer to observable movements or activities performed by entities that may involve interaction with objects or the environment in the video.
**Direction**: Direction refers to the course or path along which objects or subjects move in the video.
**Order**: Order refers to the sequential arrangement of events that occur in the video.
**Object**: Objects refer to inanimate, physical entities or items present within the video.
**State**: State refers to the condition or status of an object or subject, indicating its current properties, position or the phase of action the subject is taking or phase of process the object is undergoing.
**Count**: Count refers to the frequency of an action being performed or an event occurring. It may also refer to the number of objects or subjects involved in an event or interaction.
**Color**: Color refers to the hue or shade of an object or subject.
**Shape**: Shape refers to the form or outline of an object or subject.
**Size**: Size refers to the dimensions or magnitude of an object or subject.

Figure 17: (Left) Prompts used for generating the anchor caption, and (Right) in-context examples for the *State* sub-aspect.

You are given one or more questions targeted at content of a video and their corresponding answers. You are tasked with generating an appropriate and informative single line caption for the video using this information given to you. Ensure that you restrict yourself to only information present in the question-answer pairs provided. If the answers to the questions provide various types of information, concentrate on the color related to the subjects and objects in the video in your caption. Focus on providing clear and concise descriptions without using overly elaborate language.

`<metadata>`

Video description:

---

Original Caption:
1 : A red bucket of liquid goes from empty to half full.
Hallucinated Captions:
2 : A red bucket of liquid goes from empty to completely full.
3 : A red bucket of liquid goes from completely full to empty.

Original Caption:
1 : The light in the room is slowly dimming.
Hallucinated Captions:
2 : The light in the room slowly dims, then brightens again.
3 : The light in the room is slowly getting brighter.

Original Caption:
1 : The sky changes from clear to partly cloudy.
Hallucinated Captions:
2 : The sky changes from clear to completely overcast.
3 : The sky changes from partly cloudy to clear.

(a) Prompt used for generating the anchor caption from QA-based annotations.

(b) In-context examples for the *State* sub-aspect under the *Attribute* aspect.

Figure 18: In-context examples for the *Size* (Left) and *Shape* (Right) sub-aspects.

Original Caption:
1 : A boy inflates the balloon, which grows vertically.
Hallucinated Captions:
2 : A boy inflates the balloon, which grows horizontally.
3 : A boy deflates the balloon, which shrinks horizontally.

Original Caption:
1 : The bag expands in height as items are being placed inside.
Hallucinated Captions:
2 : The bag expands in width as items are being placed inside.
3 : The bag shrinks in height as items are being placed inside.

Original Caption:
1 : The size of the puddle of water is increasing.
Hallucinated Captions:
2 : The size of the puddle of water is decreasing.
3 : The size of the puddle of water remains unchanged.

---

Original Caption:
1 : A circle shaped block is placed in a wooden box.
Hallucinated Captions:
2 : A square shaped block is placed in a wooden box.
3 : A star shaped block is placed in a wooden box.

Original Caption:
1 : Cubes are transforming into cylinders.
Hallucinated Captions:
2 : Cubes are transforming into cones.
3 : Cubes are transforming into spheres.

Original Caption:
1 : The clouds form a fluffy circle in the sky.
Hallucinated Captions:
2 : The clouds form a fluffy square in the sky.
3 : The clouds form a fluffy triangle in the sky.

Figure 19: In-context examples for the *Color* (Left) and *Count* (Right) sub-aspects.

Original Caption:
1 : A leaf with holes turns green to red.
Hallucinated Captions:
2 : A leaf with holes turns from green to orange.
3 : A leaf with holes turns from yellow to orange.

Original Caption:
1 : A yellow ball bounces on the ground, and lands in the pool.
Hallucinated Captions:
2 : A red ball bounces on the ground, and lands in the pool.
3 : A blue ball bounces on the ground, and lands in the pool.

Original Caption:
1 : A stationary purple cup appears at the beginning of the video.
Hallucinated Captions:
2 : A stationary blue cup appears at the beginning of the video.
3 : A stationary green cup appears at the beginning of the video.

---

Original Caption:
1 : The man wearing a jacket performed three backflips.
Hallucinated Captions:
2 : The man wearing a jacket performed four backflips.
3 : The man wearing a jacket performed five backflips.

Original Caption:
1 : Four birds perched on the wire.
Hallucinated Captions:
2 : Five birds perched on the wire.
3 : Six birds perched on the wire.

Original Caption:
1 : One car drove down the road.
Hallucinated Captions:
2 : Two cars drove down the road.
3 : Three cars drove down the road.

strictly to the designated temporal aspect, ensuring that no other aspect of the video content is unintentionally altered.

Figure 20: In-context examples for the *Object* (Left) and *Event-Order* (Right) aspects.

Original Caption:
1 : A person puts a bottle in the bag. Then, he puts a book in the bag. Lastly, he puts a pencil case into the bag.
Hallucinated Captions:
2 : A person puts a book in the bag. Then, he puts a bottle in the bag. Lastly, he puts a pencil case into the bag.
3 : A person puts a pencil case in the bag. Then, he puts a book in the bag. Lastly, he puts a bottle into the bag.

Original Caption:
1 : The man hits another object with a bat.
Hallucinated Captions:
2 : The man hits another object with a racket.
3 : The man hits another object with a broom.

Original Caption:
1 : A man writes letters in the following order: A, V, T, Y.
Hallucinated Captions:
2 : A man writes letters in the following order: A, Y, T, V.
3 : A man writes letters in the following order: Y, T, V, A.

Original Caption:
1 : The ball bounces down the slanted plane.
Hallucinated Captions:
2 : The ball rolls down the slanted plane.
3 : The ball zigzags down the slanted plane.

Original Caption:
1 : A woman with white coat places a book on the table. She takes two vials of liquid and mixes them together.
Hallucinated Captions:
2 : A woman with white coat places a book on the table. She takes off her coat. Then, she takes two vials of liquid and mixes them together.
3 : A woman with white coat takes two vials of liquid and mixes them together. She then places a book on the table.

Original Caption:
1 : A person puts two rectangles and one circle into the bag.
Hallucinated Captions:
2 : A person puts a rectangle, a square and a circle into the bag.
3 : A person puts two squares and a circle into the bag.

Figure 21: In-context examples for the *Action* (Left) and *Direction* (Right) aspects.

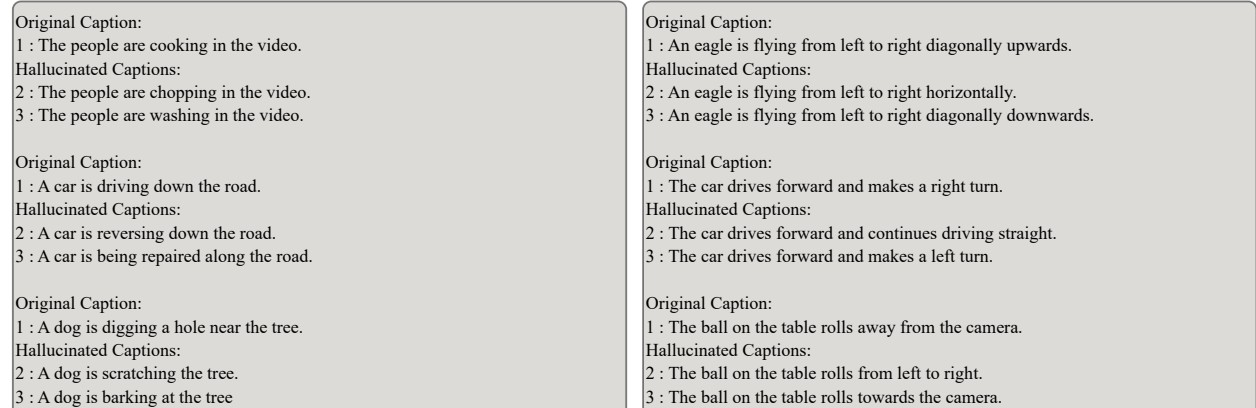

Original Caption:
1 : The people are cooking in the video.
Hallucinated Captions:
2 : The people are chopping in the video.
3 : The people are washing in the video.

Original Caption:
1 : An eagle is flying from left to right diagonally upwards.
Hallucinated Captions:
2 : An eagle is flying from left to right horizontally.
3 : An eagle is flying from left to right diagonally downwards.

Original Caption:
1 : A car is driving down the road.
Hallucinated Captions:
2 : A car is reversing down the road.
3 : A car is being repaired along the road.

Original Caption:
1 : The car drives forward and makes a right turn.
Hallucinated Captions:
2 : The car drives forward and continues driving straight.
3 : The car drives forward and makes a left turn.

Original Caption:
1 : A dog is digging a hole near the tree.
Hallucinated Captions:
2 : A dog is scratching the tree.
3 : A dog is barking at the tree

Original Caption:
1 : The ball on the table rolls away from the camera.
Hallucinated Captions:
2 : The ball on the table rolls from left to right.
3 : The ball on the table rolls towards the camera.

2. **Automated Scoring.** To assess each caption set against these criteria, we employ an ensemble of three LLMs: GPT-4o, Gemini-1.5 Flash (Reid et al., 2024), and LLaMA2 (70B) (Dubey et al., 2024) to enhance robustness against individual model biases. Each LLM independently evaluates the caption set against each criterion using binary questions, assigning a score of 1 for a positive response and 0 otherwise. Details of the prompt templates and criterion-specific questions are provided in Figures 22 and 23.

3. **Score Aggregation and Filtering.** For each criterion, the binary scores are averaged across the three LLMs in the ensemble. Instances where this average does not meet a majority consensus threshold of 0.66 (*i.e.*, where fewer than two of the three LLMs assess the criterion as satisfied) are discarded. The per-criterion scores are then summed to produce an overall quality score per instance.

**Manual Verification.** Following automated filtering, a final manual screening pass is conducted by the authors to verify that the captions and their hallucination orderings are contextually appropriate and consistent with the associated video content. Any residual misalignments, such as captions that are technically plausible but ambiguous in context, are discarded at this stage. From the remaining instances, we retain the

Figure 22: Prompt template for evaluating the quality of generated captions for the GPT-4o, Gemini-1.5 Flash, and LLaMA3 (70B) models.

---

**GPT-4o & Gemini-1.5 Flash:**
You are provided with a ground truth description of a video, and 2 other captions that contain hallucinations in the aspect of `<aspect>`. The hallucinated captions are displayed in increasing order of hallucination, where the first caption contains the least amount of hallucinated elements and the last caption having significant hallucination. You are tasked with answering a question regarding the quality of the hallucinated captions. Provide your answer as detailed in the question, without further explanation of your answer.

Ground truth caption:
`<anchor_caption>`

Hallucinated captions:
`<hallucinatory_captions>`

Question:
`<quality_assessment_question>`

Answer:

**LLaMA3 (70B):**
`<|begin_of_text|><|start_header_id|>system<|end_header_id|>`
You are provided with a ground truth description of a video, and 2 other captions that contain hallucinations in the aspect of `<aspect>`. The hallucinated captions are displayed in increasing order of hallucination, where the first caption contains the least amount of hallucinated elements and the last caption having significant hallucination. You are tasked with answering a question regarding the quality of the hallucinated captions. Provide your answer as detailed in
the question, without further explanation of your answer.
`<|eot_id|>`
`<|start_header_id|>user<|end_header_id|>`
Ground truth caption:
`<anchor_caption>`

Hallucinated captions:
`<hallucinatory_captions>`

Question:
`<quality_assessment_question>`

Answer:
`<|eot_id|>`
`<|start_header_id|>assistant<|end_header_id|>`

---

top 1,000 examples with the highest overall quality scores, while ensuring a balanced distribution of examples across temporal aspects to construct VIDHAL.

Figure 23: Question prompts for evaluating caption quality based on the three assessment criteria. Prompts with the placeholder `<option>` are applied individually to the anchor and hallucinatory captions. For question associated with *order quality*, `<option_A>` and `<option_B>` are replaced with the corresponding hallucinatory caption options shown to the LLMs.

---

**Realism:**

1. Is the scenario presented in caption `<option>` realistic? Provide your answer only as a single "yes" or "no".

2. Is the event in caption `<option>` believable? Provide your answer only as a single "yes" or "no".

3. Is the setting present in caption `<option>` plausible? Provide your answer only as a single "yes" or "no".

**Order Quality:**

1. Which caption better matches the ground truth description: Caption `<option_A>` or `<option_B>`? Provide your answer only as a single number (`<option_A>` or `<option_B>`)

2. Which caption aligns more closely with the ground truth description: Caption `<option_A>` or `<option_B>`? Provide your answer only as a single number (`<option_A>` or `<option_B>`)

3. Which caption is more faithful to the ground truth description: Caption `<option_A>` or `<option_B>`? Provide your answer only as a single number (`<option_A>` or `<option_B>`)

**Relevance:**

1. Does hallucinated caption `<option>` differ from the ground truth caption only in the `<aspect>`? Provide your answer only as a single "yes" or "no".

2. Is the only difference between hallucinated caption `<option>` and the ground truth caption the `<aspect>`? Provide your answer only as a single "yes" or "no".

3. Did hallucinated caption `<option>` change the ground truth caption only with respect to the `<aspect>`? Provide your answer only as a single "yes" or "no".

---

### A.3 Additional Dataset Examples

We provide additional qualitative examples of video instances and their corresponding captions in Figure 24 for each of the five temporal aspects.

### A.4 Hallucination Severity Criteria

Table 4 presents the operationalized criteria for moderate and high hallucination severity across all temporal aspects and sub-aspects in VIDHAL, along with illustrative caption examples for each level.

## B Human Validation Details

### B.1 Human Validation Process

As varying hallucination levels are a distinctive feature of our benchmark, we prioritize validating the robustness of caption ordering produced by our annotation pipeline. Each anchor caption is derived from the original video metadata, making it the most accurate reflection of the video content. Our primary objective is to ensure that the ordering of hallucinatory captions aligns with human judgment. To achieve this, human annotators are shown the video instance along with both hallucinatory captions and are tasked with selecting the caption that better aligns with the video content, as illustrated in Figure 25. Each video instance is reviewed by multiple annotators, with the final human-aligned order determined through a majority vote and compared with our automatically generated order.

### B.2 Analysis of Misaligned Instances

Table 5 lists video instances that fail to meet the majority agreement threshold established by our annotation process along with their corresponding human agreement scores. To assess the impact of disagreement samples on the VIDHAL evaluation suite, we evaluated several models on these instances using the full evaluation protocol. Table 6 presents the results, demonstrating that model performance on disagreement samples closely aligns with their performance on the complete benchmark, indicating that these instances do not adversely affect the overall evaluation. Upon manual inspection, we found that these disagreement cases

Figure 24: Qualitative examples of video instances and their corresponding generated captions in the VIDHAL Benchmark, across the five temporal aspects.

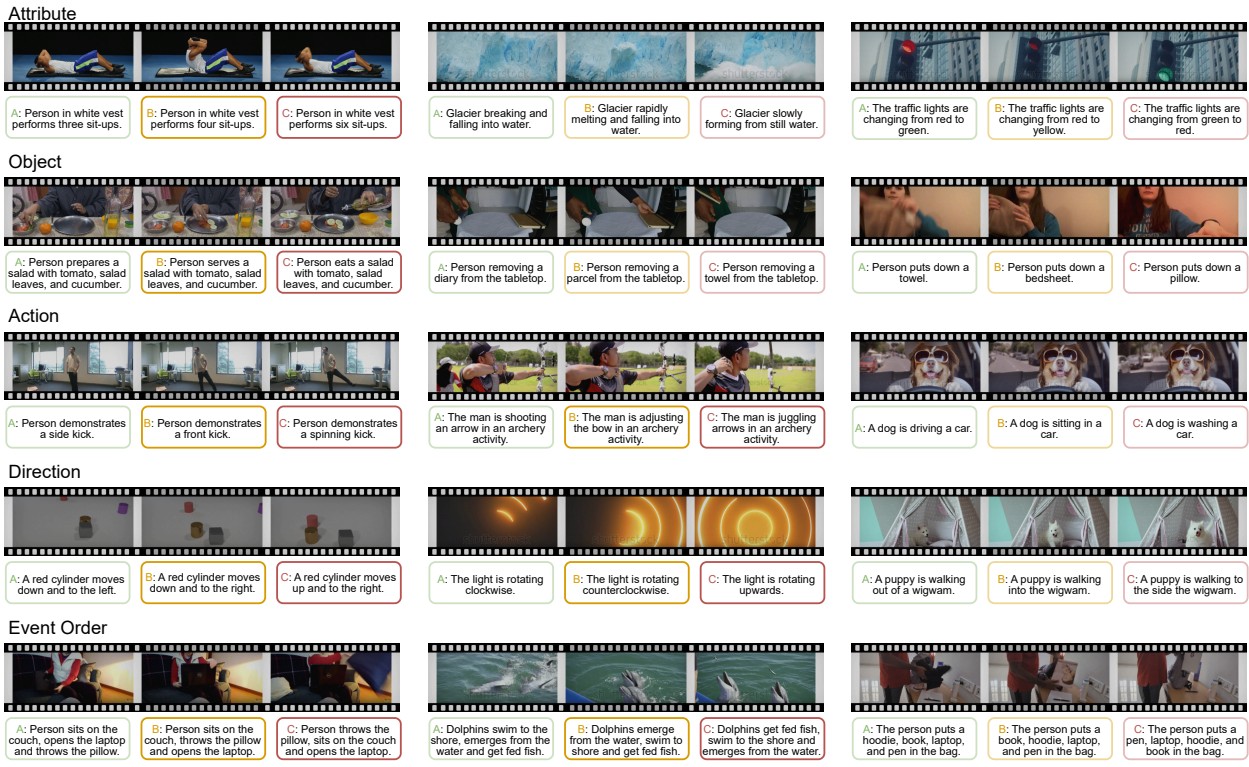

Figure 25: Pipeline for validating the quality of generated caption orders in VidHal. For each instance, human annotators are provided with the video and its associated hallucinatory captions. The annotators then select the caption that best aligns with the video content. The selected response is subsequently checked for consistency with the caption with lower hallucination according to our annotation process.

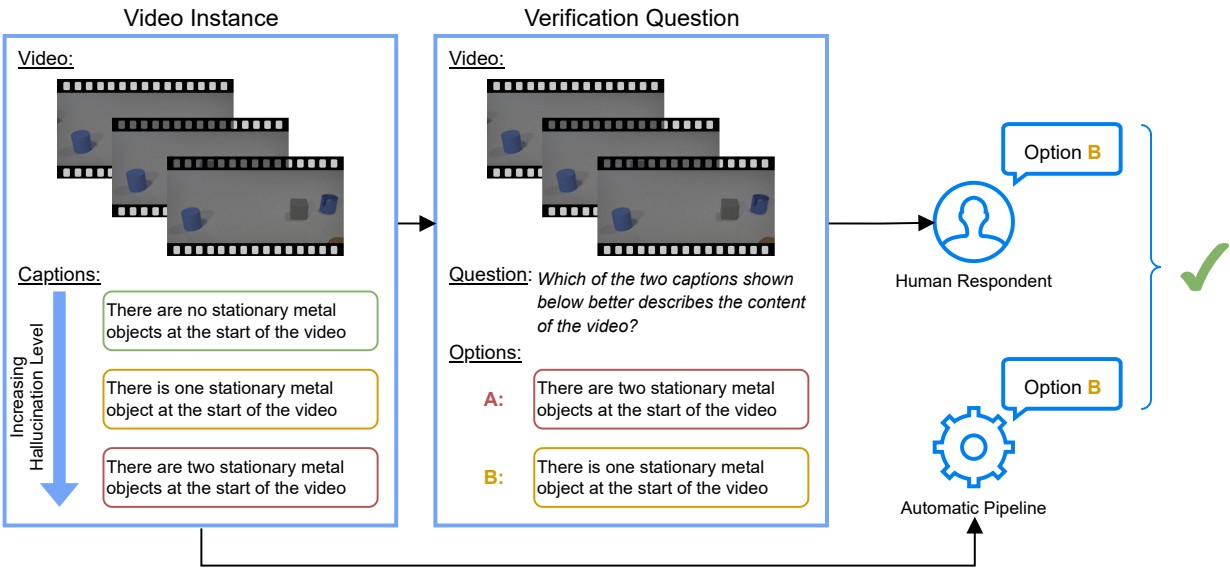

predominantly involve visually complex scenarios that are challenging even for some human annotators to

Table 4: Operationalized criteria for moderate and high hallucination severity levels across all temporal aspects and sub-aspects in VidHal, with illustrative examples in ascending order of severity.

| Aspect | Sub-aspect | Moderate Severity (MS) | High Severity (HS) | Examples (GT → MS → HS) |
|---|---|---|---|---|
| Action | — | The depicted action shares similar dynamics or movement patterns with the ground truth and remains plausible within the video context, but does not accurately represent the actual action. | The depicted action shares little to no similarity in dynamics with the ground truth and is inconsistent with the video context. | *A person is kicking a ball. → A person is throwing a ball. → A person is juggling a ball.* |
| Direction | — | The described trajectory deviates slightly from the ground truth while preserving the coarse axis of movement. | The described trajectory is diametrically opposed to or entirely deviates from the ground truth axis of movement. | *The ball rolls diagonally downward to the left. → The ball rolls diagonally downward to the right. → The ball rolls diagonally upward to the right.* |
| Object | Object Recognition | The identified object shares spatial features (e.g., shape, color, texture) with the ground truth but belongs to a distinct semantic category. | The identified object differs from the ground truth in both spatial features and semantic category. | *A man folds a blanket. → A man folds a shirt. → A man folds a paper plane.* |
| | Interaction Classification | The described interaction shares similar movements and dynamics with the ground truth but does not accurately reflect the nature of engagement between entities. | The described interaction is categorically distinct from the ground truth and inconsistent with the depicted scenario. | *The person pushes the box across the floor. → The person kicks the box. → The person throws the box.* |
| Attribute | Color | The described color is perceptually adjacent to the ground truth and may be derived from it. | The described color is perceptually distant from the ground truth and bears no resemblance to it. | *Traffic changes from green to red. → Traffic changes from blue to red. → Traffic changes from red to blue.* |
| | Shape | The described shape is geometrically similar to the ground truth, differing by at most one side. | The described shape is geometrically distinct, differing by at least two sides. | *A hexagonal block is placed in the box. → A pentagonal block. → A triangular block.* |
| | Size | The described size change deviates from the ground truth in one dimension while remaining accurate in another. | The described size change deviates in both the direction and dimension of change. | *The bag expands in height. → The bag expands in width. → The bag shrinks in width.* |
| | Count | The described count differs from the ground truth by exactly one. | The described count differs from the ground truth by at least two. | *Three birds on the wire. → Four birds. → Six birds.* |
| | State Change | The described state transition shares the same starting or ending condition as the ground truth but differs in one dimension. | The described state transition is opposite to or entirely inconsistent with the ground truth. | *The bucket goes from empty to half full. → Empty to completely full. → Completely full to empty.* |
| Event Order | — | At least one pair of events retains the correct pairwise relative ordering, while at least one other pair has an incorrect ordering. | The ordering of every pair of events is completely reversed relative to the ground truth. | *Picks up book, places in bag, zips bag. → Places book in bag, picks it up, zips bag. → Zips bag, places book in it, picks up book.* |

verify. Such difficult cases serve as valuable probes for detecting fine-grained hallucinations and distinguishing between state-of-the-art models under perceptually demanding conditions.

## C   Evaluation Pipeline Details

### C.1   Model and Inference Hyperparameters

We provide additional details on the inference and generation settings used across all evaluated models in Table 7, as well as hyperparameters specific to LlaVA-NeXT-Video models in Table 8.

## C.2   Evaluation Task Prompts

Figures 26 and 27 present the prompts used for the MCQA and naive caption ordering tasks, respectively. The same prompt used for both the MCQA task and the paired questions in the relative caption ordering task. Our manual inspection of these instances reveals that these videos often feature visually complex content, making them challenging even for human annotators.

## C.3   Relative Order Parsing

Prompting the VLLM to predict the order of captions based on their hallucinatory level in the relative caption ordering task involves asking a series of paired questions derived from different caption combinations. However, providing the model with all possible pairs at once may result in cyclic and non-transitive orderings. To address this, we present each caption pair to the VLLM in a systematically selected sequence, beginning with two paired questions. The final paired question is presented to the model to resolve inconsistencies if the multiple possible orderings can be derived from the responses to the first two paired questions. The responses across all paired questions presented to the VLLM is then parsed according to the workflow illustrated in Figure 28.

Table 5: Instances where generated caption orders diverge from human preference in quality checks. The agreement score reflects the proportion of respondents who chose our annotated order.

| Video ID | Agreement Score |
|---|---|
| action_55 | 0.429 |
| action_88 | 0 |
| action_90 | 0.308 |
| action_118 | 0.200 |
| action_153 | 0.250 |
| order_60 | 0.500 |
| order_109 | 0.154 |
| attribute_90 | 0.400 |
| attribute_180 | 0.071 |
| attribute_192 | 0.188 |
| object_25 | 0.375 |
| object_170 | 0 |
| direction_188 | 0.400 |

Figure 26: Prompt template for the MCQA and relative caption ordering evaluation tasks.

You are provided with a video and a set of several captions. Your task is to watch the video provided carefully, and select the caption that best describes the video. Provide your answer only as a single letter representing the option whose caption that best describes the video, without any explanation.

Watch the video provided, and choose the option whose caption describes the video most accurately.

A. `<caption_A>`
B. `<caption_B>`

Figure 27: Prompt template for the naive caption ordering evaluation task.

Watch the video provided, and rank the captions below in order from the most accurate to the least accurate in describing the video. Provide your response only as a sequence of comma separated option letters matching the corresponding captions. Do not give any additional explanation for your answer.

For example, if option B contains the caption that best describes the video, option A contains the caption that describes the video second best and option C contains the caption that describes the video least accurately, provide your response as: B, A, C.

A. `<caption_A>`
B. `<caption_B>`
C. `<caption_C>`

Table 6: Performance of VLLMs on disagreement samples from VIDHAL.

| Model | Accuracy | NDCG | |
|---|---|---|---|
| | | Naive | Relative |
| InternVL2.5 (8B) | 0.769 | 0.491 | 0.816 |
| Qwen2.5-VL (7B) | 0.923 | 0.814 | 0.777 |
| Gemini-2.5. (Flash) | 0.846 | 0.931 | 0.834 |

Table 7: Hyperparameter configuration used in VIDHAL evaluation across all models.

| Hyperparameter | Value |
|---|---|
| *Data Processing* | |
| Video Sampling Rate (FPS) | 30 |
| *Generation* | |
| do_sample | False |
| temperature | 0.0 |
| repetition_penalty | 1.0 |
| max_new_tokens | 128 |
| *Computation* | |
| Precision | FP16 |

Table 8: Model-specific hyperparameters for LLaVA-NeXT-Video models.

| Hyperparameter | LLaVA-NeXT-Video (7B) | LLaVA-NeXT-Video (32B) |
|---|---|---|
| mm_spatial_pool_mode | average | average |
| mm_newline_position | no_token | grid |
| mm_pooling_position | after | after |

# D  Additional Experiments

## D.1  Input Order Sensitivity

To assess the robustness of VLLM responses to the order of displayed captions, we conducted additional experiments by evaluating three VLLMs using a fixed static display order across all instances. We repeated this process across all different permutations of input caption order, presenting the results of these models in Figure 29. We observe that the performance of these VLLMs is highly sensitive to the order in which captions are displayed, reflected by their varying results across different order permutations. This instability intensifies with smaller model sizes, with VideoLLaMA2 (7B) showing the highest variance in evaluation results and VideoLLaMA2 (72B) the lowest. Our findings suggest that VLLMs may be particularly vulnerable to input caption order, potentially confounding their performance.

## D.2  Naive Caption Ordering Response Quality

To analyze VLLMs' ability to handle naive caption ordering tasks, which possess unique task structures compared to conventional video understanding tasks, we employ two quantitative metrics. Regurgitation Rate (RR) captures the model's propensity to consistently generate identical responses regardless of input, defined as the maximum proportion of instances in VIDHAL where a specific caption order is predicted across all possible orderings. Invalid Response Rate (IRR) measures the proportion of responses that fail to provide valid caption orders for the naive ordering task. Figure 30 presents IRR and RR scores for all evaluated models, revealing two key observations. First, many models exhibit high IRR scores, frequently outputting incomplete caption orders (e.g., generating only a single option). Second, despite formulating responses with

Figure 28: Decision tree for determining the final caption order based on VLLM responses to paired questions in the relative caption ordering evaluation task.

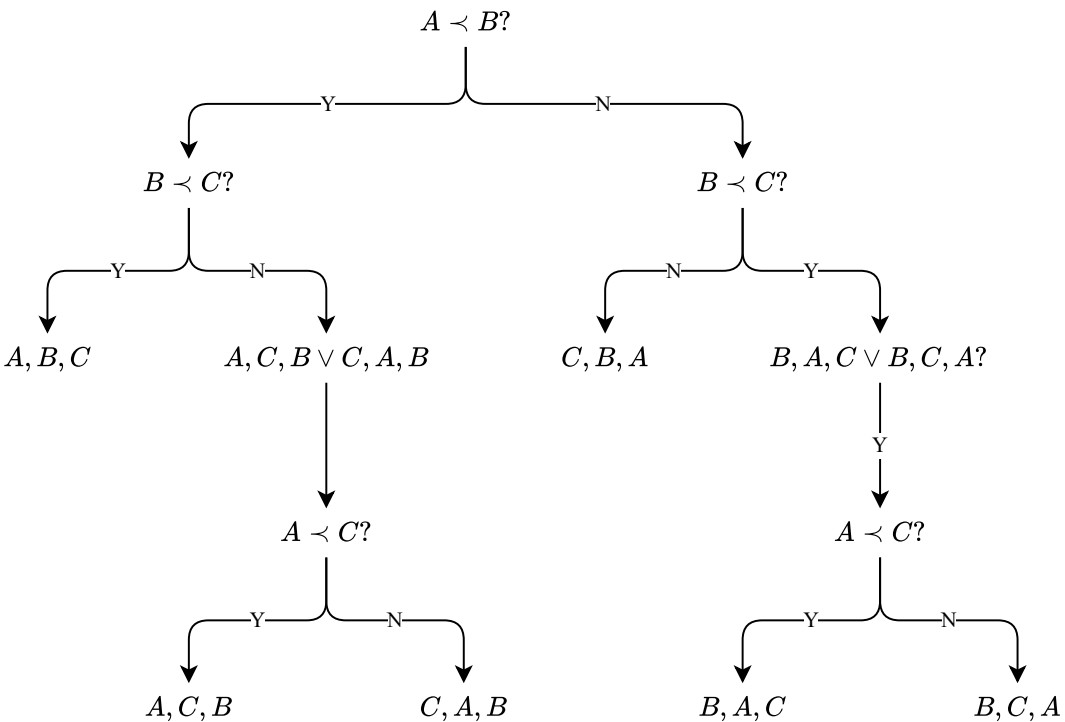

Figure 29: Distribution of results of VLLMs across varied input caption orders for the three evaluation tasks.

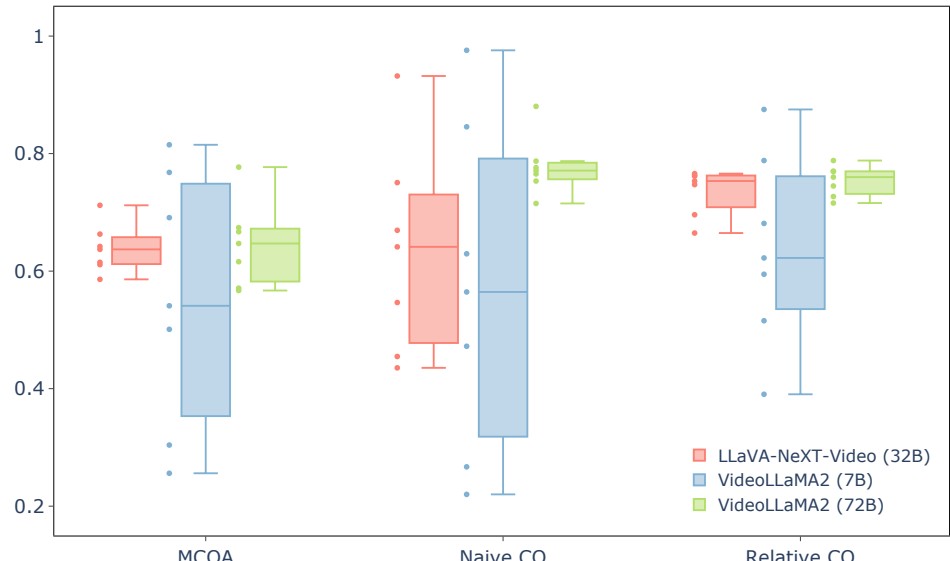

correct structure, many VLLMs produce identical caption orders regardless of the input video $V^i$, as reflected

by high RR scores, a behavior observed even in models performing well on MCQA and relative caption ordering tasks, such as InternVL2.5.

Figure 30: (Top) Invalid response rates across all models. VLLMs with no invalid responses are grouped under *Others*. (Bottom) Regurgitation rates of VLLMs on VidHal. *Random* and *Dataset Statistic* indicate the regurgitation rates of the random baseline and ground truth answers, respectively. For both metrics, a lower value indicates better model performance.

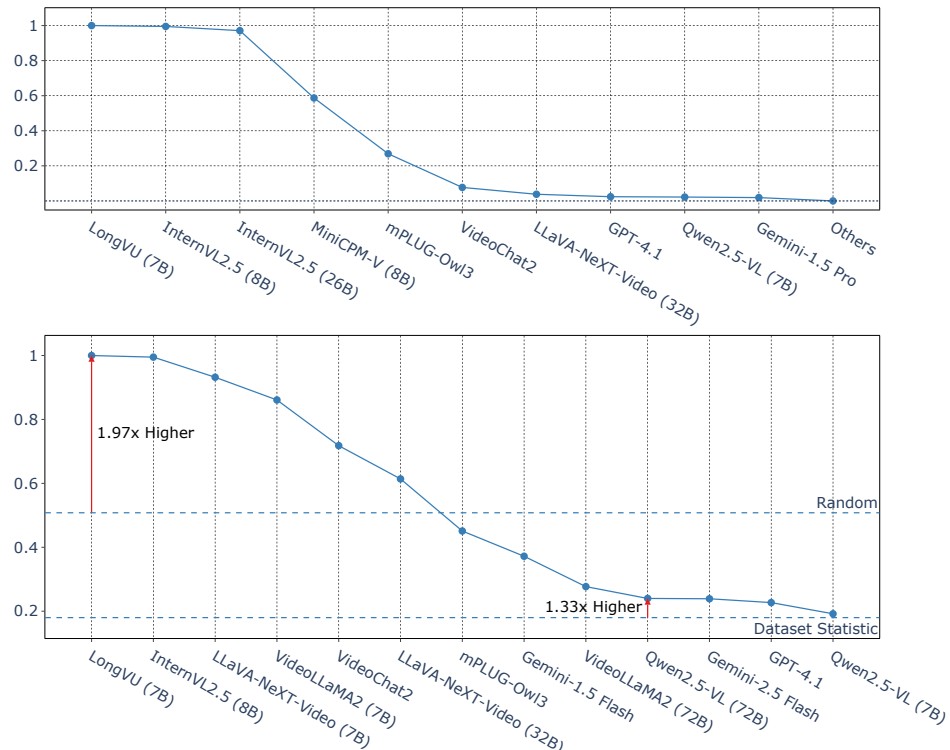

## D.3  Image Prior Reliance - Ablation Study on Video Summarization Algorithm

We conduct additional single-frame bias experiments using uniform and motion-based sampling strategies with varying clip lengths (1, 2, and 4 frames), with results presented in Tables 9 and 10. The overlap ratios demonstrate consistency across all three video summarization methods (saliency-based, uniform, and motion-based sampling) for extracting frames $v^i$. In particular, single-frame outputs substantially overlap with full-video inputs regardless of the summarization algorithm employed. These additional results confirm that our single-frame bias study is robust across different frame selection methods, with VLLMs relying on single-frame information for over half of the queries in VidHal.

Table 9: Overlapping ratios of model predictions under single-frame and full-video inputs for (C)orrect, (I)ncorrect and (O)verall predictions using uniformly sampled frames $v^i$, across multiple frame sampling rates.

| Model | 1 Frame | | | 2 Frames | | | 4 Frames | | |
|---|---|---|---|---|---|---|---|---|---|
| | **C** | **I** | **O** | **C** | **I** | **O** | **C** | **I** | **O** |
| VideoLLaMA2 (7B) | 0.674 | 0.708 | 0.700 | 0.781 | 0.798 | 0.794 | 0.846 | 0.829 | 0.833 |
| LLaVA-NeXT-Video (32B) | 0.680 | 0.570 | 0.620 | 0.735 | 0.649 | 0.688 | 0.831 | 0.706 | 0.763 |

Table 10: Overlapping ratios of model predictions under single-frame and full-video inputs for (C)orrect, (I)ncorrect and (O)verall predictions using motion-based sampled frames $v^i$, across multiple frame sampling rates.

| Model | 1 Frame | | | 2 Frames | | | 4 Frames | | |
|---|---|---|---|---|---|---|---|---|---|
| | **C** | **I** | **O** | **C** | **I** | **O** | **C** | **I** | **O** |
| VideoLLaMA2 (7B) | 0.521 | 0.495 | 0.515 | 0.558 | 0.507 | 0.519 | 0.670 | 0.653 | 0.657 |
| LLaVA-NeXT-Video (32B) | 0.634 | 0.550 | 0.558 | 0.658 | 0.546 | 0.597 | 0.675 | 0.563 | 0.614 |

### D.4 Hallucination Identification vs. Instruction Following in Naive Caption Ordering

To disentangle hallucination identification ability from instruction-following proficiency in the naive caption ordering task, we compute NDCG scores restricted to valid responses only and compare them against the overall scores reported in Table 2. A response is considered valid if the model produces a complete and parseable caption ordering. Table 11 presents both scores along with the resulting change for all evaluated models. We derive three key observations from this analysis.

- First, models with weaker instruction-following capabilities such as VideoChat2 (Mistral), mPLUG-Owl3, and MiniCPM-V 2.6 experience more substantial score increases when restricted to valid responses, reflecting the significant impact of invalid outputs on their overall NDCG.

- Second, the InternVL2.5 models exhibit disproportionately large increases attributable to near-complete invalid response rates (>95%), resulting in poor overall NDCG scores. COnsequently, their valid-response scores are computed over very few samples and are therefore subject to high variance, rendering them unreliable reflections of true hallucinatory tendencies.

- Third, and most importantly, relative rankings among top-performing models remain consistent across both evaluation settings. Leading open-source models (Qwen2.5-VL variants, VideoLLaMA2 (72B)) and proprietary models (Gemini-2.5) retain their positions, while models with notable score increases continue to lag behind by a considerable margin.

Collectively, these observations affirm the reliability of VIDHAL and the validity of the experimental insights reported in Section 2.

### D.5 Prompt Sensitivity in Naive Caption Ordering

To assess whether benchmark conclusions are robust to prompt wording, we evaluated five models across five prompt variations on the naive caption ordering task. We present the results in Table 12. Overall, stronger models exhibit greater robustness: Qwen2.5-VL (7B) shows the lowest standard deviation ($\sigma = 0.005$), while VideoLLaMA2 (7B) and LLaVA-NeXT-Video (7B) exhibit progressively higher variance consistent with their weaker overall performance. Although InternVL2.5 models also display low variance, this stems from poor instruction-following rather than genuine robustness, as evidenced by their high invalid response rates in Figure 30. With the exception of a minor ranking shift at the lower end of the performance spectrum where LLaVA-NeXT-Video (7B) falls below InternVL2.5 in some prompt variants, all relative rankings remain consistent across prompts, confirming the robustness of our benchmark conclusions to prompt formulation. The full prompt variations used in this experiment are detailed in Table 13.

Table 11: Naive caption ordering NDCG scores under overall and valid-response-only evaluation settings. Rows marked in bold (InternVL2.5) are flagged due to near-complete invalid response rates (>95%), making their valid-response scores unreliable.

| Model | NDCG (Overall) | NDCG (Valid) | Score Change |
|---|---|---|---|
| *Open-Source Models* | | | |
| VideoChat | 0.475 | 0.485 | +0.010 |
| VideoChat2 | 0.486 | 0.526 | +0.040 |
| VideoChat2 (Mistral) | 0.503 | 0.630 | +0.127 |
| VideoChat2 (Phi) | 0.626 | 0.691 | +0.065 |
| **InternVL2.5 (26B)** | **0.475** | **0.917** | **+0.442** |
| **InternVL2.5 (8B)** | **0.498** | **0.726** | **+0.228** |
| LLaVA-NeXT-Video (32B) | 0.641 | 0.667 | +0.026 |
| LLaVA-NeXT-Video (7B) | 0.518 | 0.518 | — |
| MiniCPM-V 2.6 | 0.530 | 0.684 | +0.154 |
| mPLUG-Owl3 (7B) | 0.479 | 0.655 | +0.176 |
| Qwen2.5-VL (32B) | 0.811 | 0.811 | — |
| Qwen2.5-VL (7B) | 0.825 | 0.833 | +0.008 |
| Qwen2.5-VL (72B) | 0.807 | 0.808 | +0.001 |
| VideoLLaMA2 (72B) | 0.787 | 0.790 | +0.003 |
| VideoLLaMA2 (7B) | 0.564 | 0.564 | — |
| *Proprietary Models* | | | |
| Gemini-1.5 (Flash) | 0.738 | 0.740 | +0.002 |
| Gemini-1.5 (Pro) | 0.765 | 0.779 | +0.014 |
| Gemini-2.5 (Flash) | 0.875 | 0.876 | +0.001 |
| Gemini-2.5 (Pro) | 0.876 | 0.876 | — |
| GPT-4.1 | 0.845 | 0.845 | — |
| GPT-4o | 0.840 | 0.840 | — |

Table 12: Prompt sensitivity results on the naive caption ordering task (NDCG) across five prompt variations.

| Model | Prompt 1 | Prompt 2 | Prompt 3 | Prompt 4 | Prompt 5 | Mean | Std Dev |
|---|---|---|---|---|---|---|---|
| InternVL2.5 (26B) | 0.498 | 0.546 | 0.557 | 0.474 | 0.574 | 0.530 | 0.037 |
| InternVL2.5 (8B) | 0.475 | 0.512 | 0.501 | 0.471 | 0.551 | 0.502 | 0.029 |
| LLaVA-NeXT-Video (7B) | 0.518 | 0.550 | 0.366 | 0.439 | 0.377 | 0.450 | 0.074 |
| Qwen2.5-VL (7B) | 0.825 | 0.830 | 0.836 | 0.839 | 0.834 | 0.833 | 0.005 |
| VideoLLaMA2 (7B) | 0.564 | 0.647 | 0.606 | 0.642 | 0.527 | 0.597 | 0.046 |

Table 13: Prompt variations used in the naive caption ordering prompt sensitivity study. Each prompt consists of a system instruction (S), a main task instruction (M), and a formatting hint (H). Prompt 1 indicates the original prompt used in the experiments in our paper.

| Prompt | System Instruction (S) | Main Instruction (M) | Formatting Hint (H) |
|---|---|---|---|
| 1 | You are provided with a video and a set of several captions. Your task is to order the captions from most to least relevant based on their alignment with the video content. Provide your answer without any further explanation. | Watch the video provided, and rank the captions below from the most accurate to the least accurate in describing the video. Provide your response only as a sequence of comma-separated option letters. Do not include any additional explanation in your answer. | For example, if option B best describes the video, option A describes it second best, and option C describes it least accurately, provide your response as: B, A, C. |
| 2 | You are given a video and a set of candidate captions. Your role is to evaluate how accurately each caption reflects the video content and rank the captions accordingly, from best to worst. Return only your ranking with no accompanying explanation. | Having viewed the video, rank the captions below in descending order of accuracy. Your response should consist solely of a comma-separated sequence of option letters, from the most to the least accurate caption. No further elaboration is required. | For instance, if option A is the most accurate, option C is second, and option B is least accurate, your response should be: A, C, B. |
| 3 | You are a multimodal evaluation system. Your objective is to assess the degree of descriptive alignment between a set of candidate captions and the visual content of a provided video, producing a ranked ordering from highest to lowest alignment. | Upon viewing the video, rank the captions presented below according to their descriptive accuracy relative to the observed content. Your response must be a comma-separated sequence of option identifiers ordered from the most to the least accurate. No supplementary explanation is to be included. | By way of illustration, a response indicating option A as most accurate, option C as second, and option B as least accurate would be expressed as: A, C, B. |
| 4 | You will be presented with a video alongside several descriptive captions. Your objective is to determine the extent to which each caption correctly characterises the video, and to produce a ranking from the most to the least accurate. Provide only the ranking in your response. | After watching the video, arrange the captions below in order of descriptive correctness, beginning with the caption that most precisely reflects the video content and concluding with the least precise. Express your ordering as a comma-separated list of option letters, without any additional commentary. | To illustrate, if option C most precisely reflects the video, followed by option B and then option A, your response should read: C, B, A. |
| 5 | You are provided with a video and multiple candidate captions. Your task is to rank the captions based on their fidelity to the video content, ordering them from the most to the least faithful description. Output only the final ranking. | View the video and order the captions below from the one that most faithfully describes the video to the one that does so least faithfully. Submit your answer as a comma-separated sequence of option letters in ranked order, and include no other content in your response. | As an example, if option B is the most faithful description, option A is second, and option C is least faithful, your answer should be: B, A, C. |

