# OpenReview forum: "VidHal: Benchmarking Hallucinations in Vision LLMs"
_TMLR — Accepted by TMLR_

### Review · Reviewer_mhM3 · 2026-02-23

**Summary Of Contributions:**

This paper introduces VidHal, a benchmark for evaluating temporal hallucinations in video-based Vision Large Language Models (VLLMs). Unlike prior hallucination benchmarks, VidHal introduces graded hallucination. Each video is paired with three captions, including one accurate anchor and two hallucinations with progressively increasing levels. The benchmark spans five temporal aspects (Attribute, Object, Action, Direction, Event Order). The benchmark defines two evaluation tasks: multiple-choice question answering (MCQA) and caption ordering, with the latter evaluated using NDCG adapted from information retrieval. Experiments across 23 VLLMs yield several findings, including that model scale does not consistently reduce hallucinations and that current models struggle most with temporal-specific aspects like Direction and Event Order.

Key Strengths:
- The graded hallucination framework is a meaningful conceptual advance over the binary paradigm. Distinguishing degrees of hallucination is more informative for understanding model behavior.
- The benchmark covers five well-motivated temporal aspects, providing diagnostic granularity.
- The experimental coverage is extensive (23 models, both open-source and proprietary), and the appendix includes useful supplementary analyses (input order sensitivity, invalid response rates, regurgitation rates).

Key Weaknesses:
- The definition of hallucination severity levels is underspecified and delegated to GPT-4o without a rigorous operationalization.
- The caption ordering task conflates hallucination identification ability with instruction following ability.

**Audience:**

Yes

**Audience Explanation:**

Hallucination in VLLMs is a problem of significant practical and scientific interest, and the video domain is notably under-explored compared to image-based hallucination evaluation. The empirical finding that temporal-specific aspects (Direction, Event Order) are disproportionately challenging is informative for the community, as is the observation that model scaling does not straightforwardly reduce hallucinations.

**Broader Impact Concerns:**

In general, the paper presents no significant broader impact concerns beyond those typical of benchmark papers. The videos are sourced from existing public datasets. However, the paper involves human annotators (reportedly an average of 15 per instance for the validation study) but provides no disclosure regarding their identity or whether they received compensation. Given that human annotation is central to validating the benchmark's ground-truth orderings, this lack of transparency is a notable omission, and I would encourage the authors to explicitly include a Broader Impact Statement section according to community standards.

**Claims And Evidence:**

No

**Claims Explanation:**

There are two fundamental issues that undermine my confidence in the paper's core claims:
First, the graded hallucination framework lacks a rigorous, operationalized definition of severity levels. The distinction between moderate and high hallucination is delegated entirely to GPT-4o, with no formal criteria that would allow independent reproduction or adjudication of borderline cases. Say, for the example illustrated in Figure 2, whether "clock hands are stationary" or "clock hands are moving counter-clockwise" is more severe can be argued both ways. The human validation, while showing 87% overall agreement, uses a confirmatory rather than constructive protocol (annotators judge whether a given ordering is reasonable, rather than independently producing orderings), which inflates agreement rates. Furthermore, agreement varies substantially across temporal aspects (e.g., Direction shows notably lower agreement), suggesting that for certain categories, even humans lack consensus on severity ordering.

Second, the experimental results from the caption ordering task are substantially confounded by the model's instruction following ability. Appendix D.2 reports high invalid response rates and regurgitation rates for many models on the naive ordering task, meaning these models frequently fail to produce a valid ordering at all, or default to a fixed output regardless of input. These failures reflect limitations in instruction following rather than hallucination identification, yet they directly affect the reported NDCG scores. While the relative ordering variant partially mitigates this by decomposing the task into pairwise comparisons, the paper does not sufficiently disentangle these two factors in its analysis or conclusions.

See "requested changes" for additional, secondary issues.

**Requested Changes:**

Critical (necessary for acceptance):
1. Provide a rigorous, operationalized definition of hallucination severity levels. The paper should either (a) provide explicit, human-interpretable criteria for what constitutes moderate vs. high hallucination within each temporal aspect, with illustrative examples that demonstrate the boundary conditions are not arbitrary; or (b) adopt a constructive human annotation protocol in which multiple annotators independently rank captions (without seeing a pre-generated ordering), and report inter-annotator agreement metrics to empirically establish that humans share a consistent notion of severity.
2. Disentangle hallucination identification ability from instruction following ability in the ordering task. The paper should explicitly analyze and report how much of the variance in ordering performance is attributable to instruction following failures (invalid responses, regurgitation) versus genuine differences in hallucination identification. For example, the paper could report NDCG scores computed only over valid responses, alongside the current scores, and discussing any rank changes.

Recommended (would strengthen the work)
1. Discuss the gap between discriminative evaluation and generative hallucination. VidHal evaluates whether models can identify hallucinations in externally provided captions, but the practical concern is whether models generate hallucinated content in their own outputs. The paper should include a discussion of this gap, acknowledging it as a limitation, and refer to empirical evidence from related works that justifies using the discriminative approach as a valid proxy.
2. Test prompt sensitivity. Model performance on both MCQA and ordering tasks may be sensitive to prompt wording and formatting. The paper could test 2–3 prompt variations and report the variance in model rankings.

---

> ### Author Response · Authors · 2026-04-02
> **Response to Reviewer mhM3 (1)**
>
> Thank you for the valuable feedback on our work. We provide our responses to the concerns raised below.
>
> - *Provide a rigorous, operationalized definition of hallucination severity levels.*
>
>   We thank the reviewer for this thoughtful and constructive comment. We address this concern from three aspects.
>
>   - First, we apologize for the ambiguity surrounding the human-interpretable criteria. In our approach, captions at varying hallucination levels are primarily guided by manually crafted in-context examples. As illustrated in Figures 17–21, severity is operationalized in terms of the likelihood of confusing the ground-truth aspect value with an alternative, with aspect-specific criteria summarized in the table below. Under this framework, GPT-4o is guided to generate captions by closely adhering to these examples, thereby reducing arbitrariness.
>
>   - Secondly, regarding the human annotation protocol (b), we appreciate this suggestion and wish to clarify that our existing human validation study (Section 3.4) already partially addresses this concern. Specifically, annotators were shown videos and hallucinatory captions and independently selected the caption better aligned with the video content, **without prior exposure to the pipeline-generated ordering**. The majority-vote outcome is then used to determine pairwise hallucination severity levels, which are subsequently compared against the corresponding severity levels generated by our proposed pipeline.
>
>   - Lastly, following existing majority benchmark construction studies such as [1], the caption generation is delegated to a strong model such as GPT-4o. To mitigate potential biases from over-alignment to GPT-4o preferences, we apply a two-stage quality control process comprising automated filtering using an ensemble of LLMs (Gemini-1.5, GPT-4o, and LLaMA2-70B), followed by manual verification and editing to correct any residual misalignments.
>
>   We will incorporate these criteria, along with illustrative examples for each aspect, into the revised manuscript.

---

> ### Author Response · Authors · 2026-04-02
> **Response to Reviewer mhM3 (2)**
>
> **Table: Operationalized criteria for moderate and high hallucination severity levels, with examples in ascending order of severity. GT denotes the ground-truth caption.**
>
> | Aspect | Sub-aspect | Moderate Severity (MS) | High Severity (HS) | Examples (GT → MS→ HS) |
> |---|---|---|---|---|
> | Action | — | The depicted action shares similar dynamics or movement patterns with the ground truth and remains plausible within the video context, but does not accurately represent the actual action. | The depicted action shares little to no similarity in dynamics with the ground truth and is inconsistent with the video context. | *A person is kicking a ball.* → *A person is throwing a ball.* → *A person is juggling a ball.* |
> | Direction | — | The described trajectory deviates slightly from the ground truth while preserving the coarse axis of movement. | The described trajectory is diametrically opposed to or entirely deviates from the ground truth axis of movement. | *The ball rolls diagonally downward to the left.* → *The ball rolls diagonally downward to the right.* → *The ball rolls diagonally upward to the right.* |
> | Object | Object Recognition | The identified object shares spatial features (e.g., shape, color, texture) with the ground truth but belongs to a distinct semantic category. | The identified object differs from the ground truth in both spatial features and semantic category. | *A man folds a blanket.* → *A man folds a shirt.* → *A man folds a paper plane.* |
> | Object | Interaction Classification | The described interaction shares similar movements and dynamics with the ground truth but does not accurately reflect the nature of engagement between entities. | The described interaction is categorically distinct from the ground truth and inconsistent with the depicted scenario. | *The person pushes the box across the floor.* → *The person kicks the box across the floor.* → *The person throws the box across the floor.* |
> | Attribute | Color | The described color is perceptually adjacent to the ground truth and may be derived from it. | The described color is perceptually distant from the ground truth and bears no resemblance to it (e.g., orange vs. blue). | *The traffic light changes from green to red.* → *The traffic light changes from blue to red.* → *The traffic light changes from red to blue.* |
> | Attribute | Shape | The described shape is geometrically similar to the ground truth, differing by at most one side. | The described shape is geometrically distinct from the ground truth, differing by at least two sides (e.g., triangle vs. hexagon). | *A hexagonal block is placed in the box.* → *A pentagonal block is placed in the box.* → *A triangular block is placed in the box.* |
> | Attribute | Size | The described size change deviates from the ground truth in one dimension while remaining accurate in another. | The described size change deviates from the ground truth in both the direction and dimension of change. | *The bag expands in height as items are placed inside.* → *The bag expands in width as items are placed inside.* → *The bag shrinks in width as items are placed inside.* |
> | Attribute | Count | The described count differs slightly from the ground truth by exactly one. | The described count differs greatly from the ground truth by at least two. | *Three birds are perched on the wire.* → *Four birds are perched on the wire.* → *Six birds are perched on the wire.* |
> | Attribute | State Change | The described state transition shares the same starting or ending condition as the ground truth but differs in one dimension. | The described state transition is opposite to or entirely inconsistent with the ground truth. | *The bucket goes from empty to half full.* → *The bucket goes from empty to completely full.* → *The bucket goes from completely full to empty.* |
> | Event Order | — | At least one pair of events retains the correct pairwise relative ordering from the ground truth, while at least one other pair has an incorrect sequential ordering. | The ordering of every pair of events is completely reversed relative to the ground truth, yielding the inverse event sequence. | *Person picks up a book, places it in the bag, then zips the bag.* → *Person places the book in the bag, picks it up, then zips the bag.* → *Person zips the bag, places the book in it, then picks up the book.* |

---

> ### Author Response · Authors · 2026-04-02
> **Response to Reviewer mhM3 (3)**
>
> - *Disentangle hallucination identification ability from instruction following ability in the ordering task.*
>
>   We thank the reviewer for this insightful and constructive comment. Following the suggestion, we compute NDCG scores for the naive caption ordering task restricted to valid responses only, with results compared against overall scores in the table below. We derive three key observations from this analysis.
>
>   - First, while many models exhibit modest score changes, those with weaker instruction-following capabilities, such as VideoChat2 (Mistral), mPLUG-Owl3, and MiniCPM-V 2.6, experience more substantial increases, reflecting the impact of invalid responses on their overall scores.
>   - Second, the InternVL2.5 models exhibit disproportionately large score increases, attributable to their near-complete invalid response rates (>95%) yielding poor overall NDCG scores. Consequently, their valid-response NDCG scores are computed over very few samples and may be subject to high variance, rendering them unreliable as reflections of true hallucinatory tendencies.
>   - Third, and most importantly, the relative rankings among top-performing models remain consistent across both evaluation settings. Leading open-source models (Qwen2.5-VL variants, VideoLLaMA2 (72B)) and proprietary models (Gemini-2.5) retain their positions, while models with notable score increases, such as mPLUG-Owl3 and MiniCPM-V 2.6, continue to lag behind by a considerable margin.
>
>   Collectively, these observations suggest that overall model performance and rankings remain largely stable when evaluation is restricted to valid responses, affirming the reliability of VidHal and the validity of the experimental insights reported in our paper.
>
>   **Table: Naive caption ordering NDCG scores under overall and valid-response-only evaluation. Bold rows (InternVL2.5) are flagged due to near-complete invalid response rates (>95%), making their valid-response scores unreliable.**
>
>   | Model | NDCG (Overall) | NDCG (Valid Responses) | Score Change |
>   |---|---|---|---|
>   | *Open-Source Models* | | | |
>   | VideoChat | 0.475 | 0.485 | +0.010 |
>   | VideoChat2 | 0.486 | 0.526 | +0.040 |
>   | VideoChat2 (Mistral) | 0.503 | 0.630 | +0.127 |
>   | VideoChat2 (Phi) | 0.626 | 0.691 | +0.065 |
>   | **InternVL2.5 (26B)** | **0.475** | **0.917** | **+0.442** |
>   | **InternVL2.5 (8B)** | **0.498** | **0.726** | **+0.228** |
>   | LLaVA-NeXT-Video (32B) | 0.641 | 0.667 | +0.026 |
>   | LLaVA-NeXT-Video (7B) | 0.518 | 0.518 | — |
>   | MiniCPM-V 2.6 | 0.530 | 0.684 | +0.154 |
>   | mPLUG-Owl3 (7B) | 0.479 | 0.655 | +0.176 |
>   | Qwen2.5-VL (32B) | 0.811 | 0.811 | — |
>   | Qwen2.5-VL (7B) | 0.825 | 0.833 | +0.008 |
>   | Qwen2.5-VL (72B) | 0.807 | 0.808 | +0.001 |
>   | VideoLLaMA2 (72B) | 0.787 | 0.790 | +0.003 |
>   | VideoLLaMA2 (7B) | 0.564 | 0.564 | — |
>   | *Proprietary Models* | | | |
>   | Gemini-1.5 (Flash) | 0.738 | 0.740 | +0.002 |
>   | Gemini-1.5 (Pro) | 0.765 | 0.779 | +0.014 |
>   | Gemini-2.5 (Flash) | 0.875 | 0.876 | +0.001 |
>   | Gemini-2.5 (Pro) | 0.876 | 0.876 | — |
>   | GPT-4.1 | 0.845 | 0.845 | — |
>   | GPT-4o | 0.840 | 0.840 | — |

---

> ### Author Response · Authors · 2026-04-02
> **Response to Reviewer mhM3 (4)**
>
> - *Discuss the gap between discriminative evaluation and generative hallucination.*
>
>   We thank the reviewer for raising this important point. We acknowledge that VidHal assesses a model's ability to identify hallucinated content in externally provided captions, rather than directly measuring hallucinations in model-generated outputs.
>
>   Our adoption of the discriminative paradigm is motivated by two factors: its alignment with established benchmarks such as POPE, AMBER, VideoHallucer, and VidHalluc, and its capacity for controlled, stable, and scalable assessment of model hallucinations. Crucially, discriminative evaluation exposes fundamental deficiencies in a model's perceptual grounding that are likely to manifest as hallucinated content during generation. A model unable to reliably identify incorrect descriptions of visual content is unlikely to avoid producing such inaccuracies in its own outputs.
>
>   Further supporting this, existing hallucination mitigation methods (OPERA, LURE, CSR) evaluated on both discriminative benchmarks and open-ended generative tasks consistently demonstrate that improvements on the former are accompanied by reductions in generative hallucinations, suggesting a positive correlation between the two.
>
>   We will explicitly acknowledge this inherent gap and limitation in the revised manuscript, alongside a discussion justifying discriminative evaluation as a valid proxy for generative hallucination.
>
> - *Test prompt sensitivity. Model performance on both MCQA and ordering tasks may be sensitive to prompt wording and formatting. The paper could test 2–3 prompt variations and report the variance in model rankings.*
>
>   We thank the reviewer for this constructive suggestion. To address this concern, we evaluated five models across five prompt variations on the naive caption ordering task, with results presented in the table below.
>
>   **Table: Prompt sensitivity results for selected models on the naive caption ordering task (NDCG).**
>
>   | Model | Prompt 1 | Prompt 2 | Prompt 3 | Prompt 4 | Prompt 5 | Mean | Std Dev |
>   |---|---|---|---|---|---|---|---|
>   | InternVL2.5 (26B) | 0.498 | 0.546 | 0.557 | 0.474 | 0.574 | 0.530 | 0.037 |
>   | InternVL2.5 (8B) | 0.475 | 0.512 | 0.501 | 0.471 | 0.551 | 0.502 | 0.029 |
>   | LLaVA-NeXT-Video (7B) | 0.518 | 0.550 | 0.366 | 0.439 | 0.377 | 0.450 | 0.074 |
>   | Qwen2.5-VL (7B) | 0.825 | 0.830 | 0.836 | 0.839 | 0.834 | 0.833 | 0.005 |
>   | VideoLLaMA2 (7B) | 0.564 | 0.647 | 0.606 | 0.642 | 0.527 | 0.597 | 0.046 |
>
>   Overall, stronger models exhibit greater robustness to prompt wording. For instance, Qwen2.5-VL (7B) demonstrates the lowest sensitivity (σ = 0.005), reflecting its stable video understanding capabilities, while VideoLLaMA2 (7B) and LLaVA-NeXT-Video (7B) show progressively higher variance consistent with their weaker performance on VidHal. Although InternVL2.5 (8B) and (26B) also display low variance, this is attributable to poor instruction-following on the naive caption ordering task rather than genuine robustness, as evidenced by their high invalid response rates in Figure 30. With the exception of a minor ranking shift at the lower end of the performance spectrum where LLaVA-NeXT-Video (7B) falls below InternVL2.5 across prompt variations, all relative rankings remain consistent, particularly among stronger models. This indicates that our benchmark conclusions are robust to prompt formulation.
>
> We will incorporate this analysis along with the prompt variations used into the final version of the paper.

---

> > ### Comment · Reviewer_mhM3 · 2026-04-02
> >
> > I thank the authors for their thoughtful rebuttal. My initial concerns have been resolved, and I am willing to support acceptance of the paper.
> >
> > To further strengthen this work, I would encourage the authors to conduct a text-only ablation experiment, to make sure that the model is actually looking at the video data, rather than discovering language shortcuts introduced by GPT-4o's generation.

---

### Review · Reviewer_byiM · 2026-03-20

**Summary Of Contributions:**

The paper introduces VidHal, a benchmark to evaluate video-to-text hallucinations in Vision Large Language Models (VLLMs). It addresses the lack of spatiotemporal evaluation in existing image-based benchmarks by considering video instances across various temporal aspects. The paper proposes and executes an automated pipeline to generate captions with varying levels of hallucination. Then experiments are conducted across several open-source and proprietary models: to test them on MCQA and caption ordering (hallucinatory level), Finally the result show gaps in current VLLM capabilities.

**Audience:**

Yes

**Audience Explanation:**

Yes. This paper is relevant for AI evaluation and benchmarking efforts as it introduces a structured benchmark for the underexplored area of temporal hallucinations in VLLMs. It is relevant for VLLM developers as it provides a set of metrics that models can be tested for, specifically the proposed caption-ordering task offers a more nuanced metric for model accuracy than existing binary question-answering formats.

**Broader Impact Concerns:**

None. The paper introduces VidHal, a benchmark designed to evaluate video-to-text hallucinations in Vision Large Language Models (VLLMs). It addresses the lack of spatiotemporal evaluation in existing image-based benchmarks by bootstrapping video instances across various temporal aspects. The benchmark utilizes an automated pipeline to generate captions with varying levels of hallucination and proposes a caption-ordering task to rank these levels. Experiments are conducted across several open-source and proprietary models, revealing gaps in current VLLM capabilities regarding temporal consistency.

**Claims And Evidence:**

No

**Claims Explanation:**

The methods section is currently lacking many details that make it hard to correctly interpret the results. Several key steps of the benchmark design process are under-defined, and it would be very important to read them before evaluating the correctness of the proposed benchmark. Please look at the revision requested question to get the exact details.

**Requested Changes:**

*Clear definition of hallucination and how this instantiation of the definition is different from past work on video captioning*

The paper lacks a clear definition of "hallucination." The term is used throughout the introduction and abstract, only receiving a formal definition in Section 3.1. The definition in Sec3.1 "Hallucinations in VLLMs occur when the model fabricates details in its responses that contradict the provided visual content" is quite broad. The paper does not sufficiently distinguish between a hallucination and a general inaccuracy in captioning, and whether that is relevant. Next, owing to this, it is unclear if previous literature covering video captioning errors is considered relevant or if "hallucination" is intended to be a distinct category.

*Methods require clarity, elaboration, justification and background literature*
- The paper categorizes temporal hallucinations into two levels: lexical semantics and clause semantics. It would be important to provide an explanation of how these two dimensions were derived and evidence to support the claim that they provide "holistic coverage." Similarly, the five aspects proposed for temporal concepts lack reasoning (ideally from prior literature) for their selection and comprehensiveness.
- The reliance on automated caption generation for creating ground-truth hallucinatory levels introduces potential for error. The paper does not adequately detail the specific checks, measures, or filtering mechanisms used to ensure the pipeline is robust against automation errors. It is unclear how the system guarantees that a caption intended to be "more hallucinatory" does not inadvertently contain less severe errors than a "lower-level" counterpart. The limitations section mentions using Gemini 1.5 and LLaMA2 to filter captions, but the specific details and reasoning for this filtering are missing from the methods section. Similarly, the "final step of manual verification and editing" is mentioned but not described in detail.
- The qualitative analysis in Section 5.4 lacks details about how the analysis was conducted, what were the formal techniques used. In such analysis it also important to mention what were the considerations or objectives going into the analysis, how many analysts were involved, how were any disagreements resolved, what specific criteria were used to identify the "notable patterns" described. This absence of process makes it difficult to interpret the takeaways.
- The ablation study in Section 5.3 is not clearly described, making it difficult to follow the logic or interpret the resulting data.
- Table 2 presents accuracy scores across models but lacks confidence intervals. Without these, it is impossible to determine the statistical significance of the performance gaps between the evaluated models.

*The human validation study on 100 examples raises several concerns*

- Agreement Rate: The "87% agreement rate" is not clearly defined. It is unclear if this refers to Inter-Rater Reliability (IRR) among the 15 annotators or agreement between the human average and the model’s intended ranking.
- Task Transparency: The paper lacks a description of the interface or the task given to the human validators, or of the makeup of the human validator pool. It is unclear if validators were only asked to rank captions or if they were also asked to identify if a hallucination existed at all. These are very key details.
- Sample Size: A sample of 100 out of 3,000 captions is relatively small. How was 100 arrived at? If the 13% disagreement rate reflects a true error rate in the benchmark, the paper does not discuss how this impacts the confidence levels of the overall results.
- Figure Description: Figure 3 lacks a detailed caption explaining exactly how "complete agreement" is calculated (e.g., does it require all 15 humans to agree with the model?).

*The limitations section is currently not comprehensive*
The limitations section should more explicitly describe how the benchmark design choices (synthetic generation, small validation set, specific model biases) restrict the generalizability of the findings or present caveats to the results. Specifically, it is also important to discuss how the 13% error rate (gleaned via human validation) can effect the results of a model tested in VidHal. For ex. is there a confidence interval the work can provide that captures the error rate in caption generation. This would be important to ensure correct usage of the benchmark by other researchers.

---

> ### Author Response · Authors · 2026-04-02
> **Response to Reviewer byiM (1)**
>
> We greatly appreciate the constructive comments the reviewer has presented, and would like to address them below.
>
> - *The paper lacks a clear definition of "hallucination." The term is used throughout the introduction and abstract, only receiving a formal definition in Section 3.1.*
>
>   We thank the reviewer for this insightful comment. We acknowledge that, similar to many prior visual hallucination studies, we do not adopt a strict definition of temporal hallucinations.
>
>   First, we note that the existing literature on hallucination in VLLMs, spanning hundreds of papers across both image- and video-based hallucination studies, does not reach a consensus on a unified definition of hallucination. In line with the majority of prior work, we adopt a compromised solution, treating hallucinations as model outputs that contradict the provided visual input. Our benchmark is thus distinguished from prior studies by its comprehensive coverage of temporal categories.
>
>   Second, we agree with the reviewer on the importance of distinguishing hallucinations from general inaccuracies, viewing hallucinations as a specific class of errors in which the model generates content that directly contradicts visually grounded evidence. This contrasts with errors arising from imperfect reasoning or contextual analysis that do not necessarily conflict with the visual content. This distinction motivates our focus on temporally verifiable aspects, such as direction, event order, and action, where contradictions with the visual content are clear and indicative of weak visual grounding.
>
>   We will revise the paper to introduce this characterization earlier and more explicitly to avoid ambiguity.
>
> - *The paper categorizes temporal hallucinations into two levels: lexical semantics and clause semantics. It would be important to provide an explanation of how these two dimensions were derived and evidence to support the claim that they provide "holistic coverage."*
>
>   We thank the reviewer for this insightful comment. We address it in two parts.
>
>   - **On lexical and clause semantics.** The two-level categorization is designed to provide holistic coverage over the granularity of temporal dynamics in videos. Lexical semantics targets **fine-grained** temporal dynamics, such as actions performed by entities, where hallucinations manifest as incorrect descriptions of individual temporal features. Clause semantics addresses **coarse-grained** temporal dynamics, capturing errors in the sequencing and ordering of multiple distinct events across the video. Together, these two levels span the full spectrum of temporal granularity, from intra-event structure to inter-event relational ordering. This decomposition is grounded in the temporal ontology of [1], which formally defines that both levels are necessary and sufficient for complete natural-language temporal description.
>
>   - **On the selection of the five temporal aspects.** Beyond their alignment with existing VLLM evaluation and hallucination studies, the five aspects were specifically chosen for their prevalence and importance in open-domain video understanding, where robust comprehension of these aspects is essential for general VLLM performance. This importance is further corroborated by independent video-based studies [2,3,4], which collectively identify these temporal-specific aspects as primary hallucination categories and critical dimensions for robust spatio-temporal reasoning in VLLMs. Crucially, our selection also deliberately extends beyond the static spatial understanding emphasized in prior benchmarks. For instance, the Attribute aspect not only tests object recognition but also evaluates whether VLLMs can track how attributes evolve dynamically over the course of a video. This combination of a diverse range of temporal aspects, each encompassing dynamic and temporally-grounded understanding, comprises the holistic coverage of temporal dynamics in VidHal.
>
>   We will clarify these motivations more explicitly in the revised manuscript.

---

> ### Author Response · Authors · 2026-04-02
> **Response to Reviewer byiM (2)**
>
> - *The reliance on automated caption generation for creating ground-truth hallucinatory levels introduces potential for error. The paper does not adequately detail the specific checks, measures, or filtering mechanisms used to ensure the pipeline is robust against automation errors.*
>
>   We thank the reviewer for this important comment. We acknowledge the potential for error in our automatic caption generation pipeline, and have noted this as a limitation in the conclusion section. To empirically validate the robustness of our pipeline, we conducted human validation experiments to assess whether the hallucination ordering produced by our pipeline aligns with human judgment.
>
>   Regarding the automated and manual verification steps, the process consists of the following key steps:
>
>   1. **Automated scoring**: Each generated caption set is evaluated by an ensemble of three LLMs (GPT-4o, Gemini-1.5 Flash, and LLaMA3-70B) individually across three criteria: Realism (how plausible the generated scenarios are), Ordering Quality (whether the hallucination ordering is appropriate), and Relevance (whether deviations from the anchor caption align with the designated aspect).
>   2. **Score aggregation**: For each criterion, a majority vote is taken across the three LLMs' binary assessments per instance.
>   3. **Filtering**: Instances where the majority vote indicates failure on any criterion are discarded.
>
>   Beyond this automated stage, a final manual screening pass is conducted to verify that both the captions and their assigned hallucination ordering are contextually appropriate and consistent with the associated video content. Further details on this procedure are provided in Appendix A.2.
>
> - *The qualitative analysis in Section 5.4 lacks details about how the analysis was conducted, what formal techniques were used.*
>
>   Thank you for this constructive feedback. We address this comment in two parts.
>
>   - Figure 8 presents selected instances drawn from the Action, Attribute, and Order aspects, each accompanied by the input video and predicted orderings of representative VLLMs. For each instance, the three captions are labelled A, B, and C in increasing order of hallucination level, with the ground truth ordering defined as A > B > C. Each VLLM is evaluated under both the naive and relative caption ordering tasks, with their predicted orderings recorded for direct comparison across task variants within the same instance. The broader objective of this analysis was to complement the quantitative results by providing deeper insight into model behaviour across varying task formats. To this end, we randomly sampled representative instances across the temporal aspects and evaluated the selected VLLMs on both task variants, comparing their behaviour within each instance. The Action, Attribute, and Order aspects were then selected for illustration in Figure 8.
>
>   - Given the ground truth ordering A > B > C, we assess reasoning strength and consistency by examining the number of adjacent swaps required to transform each predicted ordering into the optimal one. This yields two key observations. First, on model performance, stronger models such as VideoLLaMA2 (72B) and GPT-4o generally require at most one swap to reach the optimal ordering, which is consistent with their higher NDCG scores reported in Table 2. Second, and more importantly, on model stability, the predicted orderings of these stronger models under naive and relative caption ordering differ by at most one positional swap, as observed in the Attribute and Action aspects. This indicates that their spatiotemporal reasoning remains coherent and consistent across different task formats.
>
>   We acknowledge that these procedural details were insufficiently explicit in the original manuscript, and will revise Section 5.4 and the caption for Figure 8 to make the analysis protocol and interpretive criteria more transparent.

---

> ### Author Response · Authors · 2026-04-02
> **Response to Reviewer byiM (3)**
>
> - *The ablation study in Section 5.3 is not clearly described, making it difficult to follow the logic or interpret the resulting data.*
>
>   We thank the reviewer for this valuable feedback. Beyond the main benchmark results, these studies are designed to enrich the VidHal evaluation with deeper insights into VLLM failure modes during caption ordering, progressing from characterising *where* models fail, to understanding *why*, and finally validating the robustness of our evaluation protocol. Specifically, we provide additional clarification for each ablation study below.
>
>   - **Hallucination Differentiation Sensitivity.** We first characterise *where* models fail by investigating VLLM robustness in differentiating caption pairs of varying hallucination levels, beyond overall ordering performance. Concretely, we compute pairwise error rates across all caption pairs in the relative caption ordering task, finding that substituting a severely hallucinated caption with a moderately hallucinated one leads to substantially higher error rates against the anchor, particularly in stronger models. This reveals that model limitations are concentrated at finer-grained hallucination boundaries, underscoring the importance of graded hallucinations for comprehensively probing and advancing VLLM robustness.
>
>   - **Image Prior Reliance.** We next investigate a plausible underlying cause for *why* such deficiencies are observed in VLLMs, examining whether poor performance stems from models over-prioritising spatial over temporal information. By comparing responses under full-video and single-frame inputs, we find that model outputs remain largely consistent across both settings for at least half of VidHal instances — a finding robust across varied frame sampling strategies (Appendix D.3) — suggesting that the identified failures reflect genuine temporal reasoning deficiencies.
>
>   - **Transitive Robustness.** Lastly, since the relative caption ordering task decomposes ordering into sequential pairwise comparisons, we verify that cyclic preferences do not confound the evaluation scores. The proportion of cyclic instances is extremely low even for the weakest models, confirming that VidHal's evaluation protocol is robust against such artifacts and that reported performance differences faithfully reflect true model capability.
>
>   We apologise for the lack of clarity in the original manuscript and will revise Section 5.3 to include an introductory paragraph that elaborates on the motivation behind these studies and their procedures, ahead of each individual ablation study.
>
> - *Table 2 presents accuracy scores across models but lacks confidence intervals.*
>
>   We thank the reviewer for this constructive comment. While we agree that the inclusion of confidence intervals would add value to the study, we follow the majority of VLLM benchmarks [5,6,7,8] and employ greedy decoding, which deterministically produces the most likely response. Thus, the responses are invariant to sampling stochasticity across multiple runs.
>
> - *Agreement Rate: The "87% agreement rate" is not clearly defined. It is unclear if this refers to Inter-Rater Reliability (IRR) among the 15 annotators or agreement between the human average and the model's intended ranking.*
>
>   Thank you for raising this point. The 87% agreement rate refers to the inter-rater agreement *among* the validated instances. Specifically, this denotes the proportion of validation instances in which the caption ranking produced by our automatic generation pipeline matches the consensus ranking derived from human annotations, where the human-consensus ranking is determined by a majority vote among the rankings provided by individual annotators. We acknowledge that the current manuscript does not make this sufficiently explicit, and apologize for the confusion caused.
>
> - *Task Transparency: The paper lacks a description of the interface or the task given to the human validators, or of the makeup of the human validator pool.*
>
>   We thank the reviewer for raising this important point. We clarify that annotators were asked specifically to rank the two hallucinatory captions rather than identify whether hallucination existed. Regarding the validation interface, we engaged a third-party vendor to administer the annotation platform; as the vendor's implementation is proprietary, we do not disclose the interface directly. Nevertheless, we provide a detailed visual depiction of the overall validation workflow in Figure 25, with further elaboration in Section B.1 of the Appendix. We will revise the manuscript to incorporate these clarifications and provide greater detail in our description of the human validation process.

---

> ### Author Response · Authors · 2026-04-02
> **Response to Reviewer byiM (4)**
>
> - *Sample Size: A sample of 100 out of 3,000 captions is relatively small. How was 100 arrived at? If the 13% disagreement rate reflects a true error rate in the benchmark, the paper does not discuss how this impacts the confidence levels of the overall results.*
>
>   We are grateful to the reviewer for raising this concern. We first clarify that the 100 sampled instances represent 10% of the 1,000 total benchmark instances, rather than 3,000 captions — a sample size chosen to balance representativeness with annotation cost, consistent with standard benchmark construction practice. Regarding the 13% disagreement rate, we investigate its potential impact on overall benchmark results through empirical evaluation. As presented in Table 5 and discussed in Section B.2, we evaluate several VLLMs spanning both open-source and proprietary models on these disagreement samples across all three evaluation tasks. Performance on the disagreement subset closely mirrors that on the full benchmark, with no significant deviation observed across any model family or task format. This empirically supports that the disagreement instances do not introduce bias or adversely affect the validity of our conclusions.
>
> - *Figure Description: Figure 3 lacks a detailed caption explaining exactly how "complete agreement" is calculated (e.g., does it require all 15 humans to agree with the model?).*
>
>   Thank you for this thoughtful observation. The "Complete Agreement" dotted line in the right graph of Figure 3 serves as a reference marker at an agreement ratio of 1.0, indicating the theoretical upper bound at which human-consensus rankings fully align with VidHal's automatically generated rankings across all sampled instances. This label is therefore intended to denote the upper bound of the overall agreement rate over the validation instances, rather than a condition requiring unanimous consensus among all annotators. We acknowledge that this distinction was not made sufficiently clear in the original caption, and will include additional detail in the caption of Figure 3 in the revised manuscript.
>
> - *The limitations section is currently not comprehensive. The limitations section should more explicitly describe how the benchmark design choices restrict the generalizability of the findings or present caveats to the results.*
>   We thank the reviewer for this constructive feedback and would like to address this comment in two parts.
>
>   - To identify the effect of the slight disagreement rate on model performance on VidHal, we perform additional experiments on the disagreement samples across several VLLMs, with results presented in Table 5 (Appendix B.2). We find that model performance on these samples does not deviate greatly from their overall benchmark performance, suggesting that the residual disagreement introduces minimal skew to the reported results.
>   - When constructing the caption generation process, we carefully design it to ensure comprehensive coverage of various temporal aspects and semantic categories to provide a holistic assessment. Nevertheless, the hallucinations present in our synthetic captions may still deviate from the internal biases and preferences of the VLLMs being evaluated, which represents a caveat that researchers should account for when interpreting evaluation results on VidHal.
>
>   We will update the limitations section to make both of these points more explicit to readers.
>
> [1] Marc Moens and Mark Steedman. 1988. Temporal Ontology and Temporal Reference.
>
> [2] Wang et al. (2024). VideoHallucer: Evaluating Intrinsic and Extrinsic Hallucinations in Large Video-Language Models.
>
> [3] Liu et al. (2024). TempCompass: Do Video LLMs Really Understand Videos?
>
> [4] Li et al. (2024). MVBench: A Comprehensive Multi-modal Video Understanding Benchmark.
>
> [5] Li et al. (2024). VidHalluc: Evaluating Temporal Hallucinations in Multimodal Large Language Models for Video Understanding.
>
> [6] Ben-Kish et al. (2024). Mitigating Open-Vocabulary Caption Hallucinations.
>
> [7] Ma et al. (2025). VideoEval-Pro: Robust and Realistic Long Video Understanding Evaluation.
>
> [8] Fang et al. (2024). MMBench-Video: A Long-Form Multi-Shot Benchmark for Holistic Video Understanding.

---

### Review · Reviewer_gmox · 2026-03-20

**Summary Of Contributions:**

Summary:

The authors create a benchmark for evaluating hallucinations of vision LLMs in _videos_. It is based on four public datasets and contains 1,000 videos. Each video has three corresponding captions: one correct caption, and two with different degrees of hallucinations. They use their benchmark to evaluate 23 models from 13 model families, both open-source and proprietary models.




***
Strengths:
- The studied problem is interesting, I think it makes sense to specifically study _video_-based hallucinations.
- The paper is well written and easy to follow overall. The dataset construction in Section 3 is explained well, and Figure 2 gives a good overview.
- They use their benchmark to evaluate a large set of models.






***
Weaknesses:
- Results are not ideally presented in Figure 1 and Figure 5.
- The benchmark results could be discussed more, for example why small versions of InternVL2.5 and Qwen2.5-VL outperform large versions.





***
Questions/suggestions:
- I think there might be a bit too many lines and too similar colors for the radar plots in Figure 1 and 5 to be effective, I find them difficult to parse at least. I think there might be better ways to present these results. Or, show these radar plots just for a smaller subset of models, e.g. the top-performing models. Also, in Figure 5 some models have different colors in (a) vs (b)? Also, in Figure 1 (left), there is no scale or metric.
- When reading the paper, I was a bit surprised that you set M=3, I was expecting a larger number. Did you attempt generating more than just two negative captions, did this not work well?
- You have some space left to the soft page limit of 12 pages, could it not make sense to move some more details to the main paper? For example, I think Figure 11 shows interesting/important benchmark information. And, the examples in Figure 24 very clearly show what the benchmark evaluates.
- I think the results for InternVL2.5 and Qwen2.5-VL should be discussed a bit more. It's surprising that a 7B model outperforms a 72B model, right? Do you have any possible explanation for this?
- Figure 7 is interesting. The three evaluated models there are not among the best-performing models though, could be interesting to do this analysis also for e.g. Qwen2.5-VL (7B)?
- LongVU is the best among all open-source models in terms of both Accuracy and Relative in Table 2, but the very worst model in terms of Naive? Isn't this strange?
- For the "Parameter Scale vs. Performance" paragraph in Section 5.2, I think you should probably mention/clarify the results for LLaVA-NeXT-Video and VideoLLaMA2. For these, performance is improved with increased model size. But, the two different model sizes use different LLMs, this could be made more clear.





***
Minor things:
- I think table captions should be placed above tables in TMLR.
- "NDCG" and "MCQA" have not been introduced/defined when they're used in Section 1.
- I assume the dashed blue line in Figure 1 (right) is the average performance across models, but this is not described anywhere.
- Section 2, "and Vript (Yang et al., 2024) provides" --> "and Vript (Yang et al., 2024) provide"?
- Section 2, "and VideoHallucer (Wang et al., 2024b) introduces" --> "and VideoHallucer (Wang et al., 2024b) introduce"?
- Start of Section 3, "designed to evaluate hallucinations of Video-LLMs", the use of "video-LLMs" here seems inconsistent with the text prior? Or, would it make more sense to change from "Vision LLMs" to "Video LLMs" in the title etc?
- Above equation (4), "with lower hallucinatory extent." --> "with lower hallucinatory extent:"?
- 5.1, "and two proprietary models" --> "and three proprietary models"?
- Table 2, why "1fps" but "16", why not "16fps"? Or why not remove "fps" entirely?
- Equation (6), "." --> "," at the end?

**Audience:**

Yes

**Audience Explanation:**

The studied problem of _video_-based hallucinations is interesting, and I definitely think the proposed benchmark could be useful.

**Broader Impact Concerns:**

No concerns.

**Claims And Evidence:**

No

**Claims Explanation:**

I think the paper would benefit from more clearly presented results, extended discussion of the results, and a few clarifications.

**Requested Changes:**

This is a quite interesting and well-written paper that I definitely think could be relevant for the TMLR audience.

However, I think the current version would benefit from some clarifications and modifications, see "Weaknesses" and "Questions/suggestions" above.

---

> ### Author Response · Authors · 2026-04-02
> **Response to Reviewer gmox (1)**
>
> We thank the reviewer for their insightful comments and suggestions, and provide our responses to them below.
>
> - *Results are not ideally presented in Figure 1 and Figure 5.*
>
>   We thank the reviewer for their valuable feedback. In the revised manuscript, Figures 1 and 5 have been updated to display results for a representative subset of models. Additionally, Figure 5 has been revised to ensure consistent color coding across related plots, improving overall clarity and readability.
>
> - *When reading the paper, I was a bit surprised that you set M=3, I was expecting a larger number. Did you attempt generating more than just two negative captions, did this not work well?*
>
>   Thank you for your thoughtful observation, this is indeed an important design choice that warrants further explanation. We did experiment with a range of values $M \in $ {$3, 4, 5$}. With larger values of M, we encountered two key issues. First, the stability of automatic hallucinatory caption generation degrades: as M increases, the boundaries between adjacent hallucination levels become increasingly fine-grained, making them difficult to distinguish even for strong proprietary models such as GPT-4o. Second, increasing M raises the cost of human validation considerably, rendering both the annotation and verification stages prohibitively expensive at scale. As $M = 3$ produced hallucinatory captions whose relative ordering aligned well with human judgment, we adopted this value in our final design.
>
>   Furthermore, our experiments demonstrate that current VLLMs experience considerable difficulty when presented with three captions. Thus, we reserve the exploration of larger values of M as a promising direction for future extensions of VidHal, as the reasoning capabilities of VLLMs continue to improve.
>
> - *You have some space left to the soft page limit of 12 pages — could it not make sense to move some more details to the main paper?*
>
>   We would like to thank the reviewer for this observation. We will incorporate them, along with other adjustments suggested by the reviewers, to fill the remaining space within the 12-page limit.
>
> - *The benchmark results could be discussed more, for example why small versions of InternVL2.5 and Qwen2.5-VL outperform large versions. It's surprising that a 7B model outperforms a 72B model*
>
>   We thank the reviewer for raising this interesting point. We agree this is a finding that warrants more thorough discussion, and will address it more explicitly in the revised manuscript. That said, a plausible explanation lies in the asymmetric scaling between the vision encoder and the backbone LLM across model sizes. The increase in model capacity in larger variants is predominantly driven by LLM scaling rather than a proportional enhancement of the vision encoder, which may cause larger models to rely more heavily on language-based priors at the expense of fine-grained visual reasoning. This aligns with our observation in Section 5.2 that scaling model capacity alone may offer limited benefits for reducing video-based hallucinations.
>
>   **Table: Overlapping ratios of Qwen2.5-VL (7B) predictions under single-frame and full-video inputs for correct, incorrect, and overall predictions in the naive and relative caption ordering tasks.**
>
>   | Task | Correct | Incorrect | Overall |
>   |---|---|---|---|
>   | Naive | 0.785 | 0.363 | 0.554 |
>   | Relative | 0.767 | 0.322 | 0.511 |
>
>   Consistent with the findings in Figure 7, Qwen2.5-VL (7B) exhibits substantial image prior reliance, with over half of its responses remaining unchanged when given only a single frame. Notably, compared to models of similar parameter scale, it demonstrates comparatively lower overlap ratios, suggesting a greater capacity to leverage temporal information from the full video to refine its predictions. We will incorporate these results in our revised manuscript.

---

> ### Author Response · Authors · 2026-04-02
> **Response to Reviewer gmox (2)**
>
> - *LongVU is the best among all open-source models in terms of both Accuracy and Relative in Table 2, but the very worst model in terms of Naive — isn't this strange?*
>
>   We appreciate the reviewer's insightful observation. We hypothesize that while LongVU's architectural design yields strong spatiotemporal understanding, as reflected in its leading Accuracy and Relative caption ordering performance, it undergoes comparatively less instruction-following fine-tuning than models such as Qwen2.5-VL. This weaker instruction-following ability limits its capacity to correctly interpret the more demanding naive ordering format, leading to frequent invalid responses and disproportionately poor performance on this task, as evidenced by its notably high invalid response rate in Figure 30. Such behaviour is consistent with observations from our preliminary experiments during the design of our evaluation tasks, and directly motivated the introduction of both task variants. Relative caption ordering provides a fairer assessment of spatiotemporal reasoning across models of varying capability levels. Naive caption ordering, in turn, serves as an additional discriminator for identifying models with stronger instruction-following proficiency.
>
> - *For the "Parameter Scale vs. Performance" paragraph in Section 5.2, I think you should probably mention/clarify the results for LLaVA-NeXT-Video and VideoLLaMA2.*
>
>   Thank you for this careful observation. We will make this clearer in our revision: while the LLaVA-NeXT-Video and VideoLLaMA2 model families adopt different backbone LLMs across model sizes, the performance gains observed in larger variants are more likely attributable to increased parameter scale rather than the choice of LLM.
>
> Regarding the minor comments raised, the following structural and syntactic corrections have been incorporated into the revision:
>
> - Table captions repositioned above tables per TMLR guidelines.
> - Subject-verb agreement corrected for "Vript" and "VideoHallucer" in Section 2.
> - Punctuation updated above Equation (4) and at the end of Equation (6).
> - "two proprietary models" corrected to "three proprietary models" in Section 5.1.
>
> We address the remaining minor comments below:
>
> - *"I assume the dashed blue line in Figure 1 (right) is the average performance across models, but this is not described anywhere."*
>
>   The dashed blue line does indeed represent the average performance across models. We apologise for the omission of this information. To avoid further confusion and to improve the visibility of individual model performance, we will remove this line from the revised figure.
>
> - *"NDCG and MCQA have not been introduced/defined when they are used in Section 1."*
>
>   We apologise for this oversight. In the revision, we will replace these acronyms in Section 1 with more general descriptions of the corresponding tasks, reserving their formal definitions and acronym introductions for Section 4, where the evaluation protocol is presented.
>
> - *"Start of Section 3, 'designed to evaluate hallucinations of Video-LLMs' — the use of 'Video-LLMs' here seems inconsistent with the text prior."*
>
>   We apologise for the inconsistency. We have replaced all instances of "Video-LLMs" with "Vision LLMs" to maintain consistency throughout the paper.
>
> - *"Table 2, why '1fps' but '16'? Why not '16fps'? Or why not remove 'fps' entirely?"*
>
>   For most open-source models, evaluation follows the recommended settings from the original works, which typically specify a fixed number of frames rather than a frame rate. For proprietary models, as no official frame-sampling recommendations exist, we follow prior evaluation studies [1] and sample at 1 fps. To better reflect this distinction, we will revise the table to use a consistent notation that clearly differentiates between fixed frame counts and frame rates.
>
>
> [1] Fu et al. (2024). Video-MME: The First-Ever Comprehensive Evaluation Benchmark of Multi-modal LLMs in Video Analysis.

---

> > ### Comment · Reviewer_gmox · 2026-04-05
> >
> > Thank you for the detailed response.
> >
> > I have read the other reviews and rebuttals. I think the authors responded very well. I don't really see any reason why this paper shouldn't be accepted.

---

### Author Response · Authors · 2026-04-23
**Author Closing Remarks and Acknowledgments**

We sincerely thank the action editor and reviewers for their time and effort in evaluating our manuscript, as well as for their thoughtful feedback and valuable insights, which have contributed meaningfully to improving the quality and clarity of the paper. We express our gratitude for their support in bringing this work to publication.

---

### Decision · Action_Editor_fDcr · 2026-04-18

**Recommendation:** Accept as is

**Audience:**

Yes

**Audience Explanation:**

This work addresses an important and underexplored problem in evaluating video-based hallucinations in VLLMs which will be of interest and value to researchers in the area.

**Claims And Evidence:**

Yes

**Claims Explanation:**

The paper introduces a benchmark for evaluating video-based hallucinations in VLLMs using graded hallucination captions and a caption-ordering task, showing that current models struggle with temporal reasoning and hallucination severity. The rebuttal clarified some definitions (hallucination vs. general errors), strengthened some methodological details (caption generation, filtering, human validation), and added analyses for results (e.g., image-prior reliance, scaling effects), and addressed most reviewer concerns and improving confidence in the benchmark.